# IMITATING HUMAN BEHAVIOUR WITH DIFFUSION MODELS

**Tim Pearce, Tabish Rashid, Anssi Kanervisto, Dave Bignell, Mingfei Sun, Raluca Georgescu, Sergio Valcarcel Macua, Shan Zheng Tan, Ida Momennejad, Katja Hofmann, Sam Devlin**
Microsoft Research

## ABSTRACT

Diffusion models have emerged as powerful generative models in the text-to-image domain. This paper studies their application as observation-to-action models for imitating human behaviour in sequential environments. Human behaviour is stochastic and multimodal, with structured correlations between action dimensions. Meanwhile, standard modelling choices in behaviour cloning are limited in their expressiveness and may introduce bias into the cloned policy. We begin by pointing out the limitations of these choices. We then propose that diffusion models are an excellent fit for imitating human behaviour, since they learn an expressive distribution over the joint action space. We introduce several innovations to make diffusion models suitable for sequential environments; designing suitable architectures, investigating the role of guidance, and developing reliable sampling strategies. Experimentally, diffusion models closely match human demonstrations in a simulated robotic control task and a modern 3D gaming environment.
Code: https://github.com/microsoft/Imitating-Human-Behaviour-w-Diffusion.

## 1 INTRODUCTION

To enable Human-AI collaboration, agents must learn to best respond to all plausible human behaviors (Dafoe et al., 2020; Mirsky et al., 2022). In simple environments, it suffices to generate all possible human behaviours (Strouse et al., 2021) but as the complexity of the environment grows this approach will struggle to scale. If we instead assume access to human behavioural data, collaborative agents can be improved by training with models of human behaviour (Carroll et al., 2019).

In principle, human behavior can be modelled via imitation learning approaches in which an agent is trained to mimic the actions of a demonstrator from an offline dataset of observation and action tuples. More specifically, Behaviour Cloning (BC), despite being theoretically limited (Ross et al., 2011), has been empirically effective in domains such as autonomous driving (Pomerleau, 1991), robotics (Florence et al., 2022) and game playing (Ye et al., 2020; Pearce and Zhu, 2022).

Popular approaches to BC restrict the types of distributions that can be modelled to make learning simpler. A common approach for continuous actions is to learn a point estimate, optimised via Mean Squared Error (MSE), which can be interpreted as an isotropic Gaussian of negligible variance. Another popular approach is to *discretise* the action space into a finite number of bins and frame as a classification problem. These both suffer due to the approximations they make (illustrated in Figure 1), either encouraging the agent to learn an 'average' policy or predicting action dimensions independently resulting in 'uncoordinated' behaviour (Ke et al., 2020).

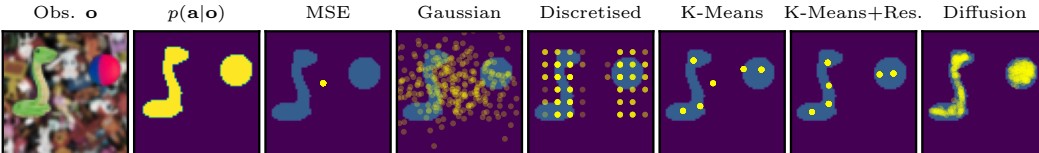

Figure 1: Expressiveness of a variety of models for behaviour cloning in a single-step, arcade claw game with two simultaneous, continuous actions. Existing methods fail to model the full action distribution, $p(\mathbf{a}|\mathbf{o})$, whilst diffusion models excel at covering multimodal & complex distributions.

Such simplistic modelling choices can be successful when the demonstrating policy is itself of restricted expressiveness (e.g. when using trajectories from a single pre-trained policy represented by a simple model.) However, for applications requiring cloning of human behaviour, which contains diverse trajectories and multimodality at decision points, simple models may not be expressive enough to capture the full range and fidelity of behaviours (Orsini et al., 2021).

For these reasons, we seek to model the full distribution of actions observed. In particular, this paper focuses on diffusion models, which currently lead in image, video and audio generation (Saharia et al., 2022; Harvey et al., 2022; Kong et al., 2020), and avoid issues of training instability in generative adversarial networks (Srivastava et al., 2017), or sampling issues with energy based models (Florence et al., 2022). By using diffusion models for BC we are able to: 1) more accurately model complex action distributions (as illustrated in Figure 1); 2) significantly outperform state-of-the-art methods (Shafiullah et al., 2022) on a simulated robotic benchmark; and 3) scale to modelling human gameplay in Counter-Strike: Global Offensive - a modern, 3D gaming environment recently proposed as a platform for imitation learning research (Pearce and Zhu, 2022).

To achieve this performance, we contribute several innovations to adapt diffusion models to sequential environments. Section 3.2 shows that good architecture design can significantly improve performance. Section 3.3 then shows that Classifier-Free Guidance (CFG), which is a core part of text-to-image models, surprisingly harms performance in observation-to-action models. Finally, Section 3.4 introduces novel, reliable sampling schemes for diffusion models. The appendices include related work, experimental details, as well as further results and explanations.

## 2 MODELLING CHOICES FOR BEHAVIOURAL CLONING

In this section we examine common modelling choices for BC. For illustration purposes, we created a simple environment to highlight their limitations. We simulated an arcade toy claw machine, as shown in Figures 1, 3 & 4. An agent observes a top-down image of toys ($\mathbf{o}$) and chooses a point in the image, in a 2D continuous action space, $\mathbf{a} \in \mathbb{R}^2$. If the chosen point is inside the boundaries of a valid toy, the agent successfully obtains the toy. To build a dataset of demonstrations, we synthetically generate images containing one or more toys, and uniformly at random pick a single demonstration action $\mathbf{a}$ that successfully grabs a toy (note this toy environment uses synthetic rather than human data). The resulting dataset is used to learn $\hat{p}(\mathbf{a}|\mathbf{o})$. To make training quicker, we restrict the number of unique observations $o$ to seven, though this could be generalised.

**MSE.** A popular choice for BC in continuous action spaces approximates $p(\mathbf{a}|\mathbf{o})$ by a point-estimate that is optimised via MSE. This makes a surprisingly strong baseline in the literature despite its simplicity. However, MSE suffers from two limitations that harm its applicability to our goal of modelling the full, complex distributions of human behaviour. 1) MSE outputs a point-estimate. This precludes it from capturing any variance or multimodality present in $p(\mathbf{a}|\mathbf{o})$. 2) Due to its optimisation objective, MSE learns the 'average' of the distribution. This can bias the estimate towards more frequently occurring actions, or can even lead to out-of-distribution actions (e.g. picking the action between two modes). The first can be partially mitigated by instead assuming a Gaussian distribution, predicting a variance for each action dimension and sampling from the resulting Gaussian. However, due to the MSE objective, the learnt mean is still the average of the observed action distribution. These limitations are visualised in Figure 1.

**Discretised.** A second popular choice is to discretise each continuous action dimension into $B$ bins, and frame it as a classification task. This has two major limitations. 1) Quantisation errors arise since the model outputs a single value for each bin. 2) Since each action dimension is treated independently, the marginal rather than the joint distribution is learnt. This can lead to issues during sampling whereby dependencies between dimensions are ignored, leading to 'uncoordinated' behaviour. This can be observed in Figure 1 where points outside of the true distribution have been sampled in the bottom-right corner. This can be remedied by modelling action dimensions autoregressively, but these models bring their own challenges and drawbacks (Lin et al., 2021).

**K-Means.** Another method that accounts for dependencies between action dimensions, first clusters the actions across the dataset into $K$ bins (rather than $B^{|\mathbf{a}|}$) using K-Means. This discretises the joint-action distribution, rather than the marginal as in 'Discretised'. Each action is then associated with its nearest cluster, and learning can again be framed as a classification task. This approach

avoids enumerating all possible action combinations, by placing bins only where datapoints exist. However two new limitations are introduced: 1) Quantisation errors can be more severe than for 'Discretised' due to only $K$ action options. 2) The choice of $K$ can be critical to performance (Guss et al., 2021). If $K$ is small, important actions may be unavailable to an agent; if $K$ is large, learning becomes difficult. Additionally, since the bins are chosen with respect to the entire dataset they can fall outside of the target distribution for a given observation. This is observed in Figure 1.

**K-Means+Residual.** Shafiullah et al. (2022) extended K-Means by learning an observation-dependent residual that is added to the bin's center and optimised via MSE. This increases the fidelity of the learnt distribution. However, it still carries through some issues of K-Means and MSE. 1) The distribution $p(\mathbf{a}|\mathbf{o})$ is still modelled by a finite number of point estimates (maximum of $K$). 2) The residual learns the 'average' of the actions that fall within each bin. In addition, it still requires a careful choice of the hyperparameter K.

**Diffusion models.** The limitations of these existing modelling choices all arise from approximations made in the form of $\hat{p}(\mathbf{a}|\mathbf{o})$, combined with the optimisation objective. But it is precisely these approximations that make training models straightforward (optimising a network via MSE or cross-entropy is trivial!). So how can we learn $p(\mathbf{a}|\mathbf{o})$ while avoiding approximations? We propose that recent progress in generative modelling with diffusion models provides an answer. Diffusion models are able to output expressive conditional joint distributions, and are powerful enough to scale to problems as complex as text-to-image or video generation. Thus, we will show they provide benefits when imitating human demonstrations as they make no coarse approximations about the action distribution, and avoid many of the limitations of the previously discussed choices. This can be observed in Figure 1, where only diffusion provides an accurate reconstruction of the true action distribution.

## 3 OBSERVATION-TO-ACTION DIFFUSION MODELS

Diffusion models have largely been developed for the image domain. This section first introduces the underlying principles of diffusion models, then studies several aspects that require consideration to apply them to sequential environments.

### 3.1 DIFFUSION MODEL OVERVIEW

Diffusion models are generative models that map Gaussian noise to some target distribution in an iterative fashion, optionally conditioned on some context (Dhariwal and Nichol, 2021). Beginning from, $\mathbf{a}_T \sim \mathcal{N}(\mathbf{0}, \mathbf{I})$, a sequence $\mathbf{a}_{T-1}, \mathbf{a}_{T-2} \dots \mathbf{a}_0$ is predicted, each a slightly denoised version of the previous, with $\mathbf{a}_0$ a 'clean' sample. Where $T$ is the total number of denoising steps (*not* the environment timestep as is common in sequential decision making).

This paper uses denoising diffusion probabilisitic models (Ho et al., 2020). During training, noisy inputs can be generated as: $\mathbf{a}_\tau = \sqrt{\bar{\alpha}_\tau}\mathbf{a} + \sqrt{1 - \bar{\alpha}_\tau}\mathbf{z}$, for some variance schedule $\bar{\alpha}_\tau$, random noise $\mathbf{z} \sim \mathcal{N}(\mathbf{0}, \mathbf{I})$. A neural network, $\epsilon(\cdot)$, is trained to predict the noise that was added to an input, by minimising:

$$\mathcal{L}_{\text{DDPM}} \coloneqq \mathbb{E}_{\mathbf{o}, \mathbf{a}, \tau, \mathbf{z}} \left[ ||\epsilon(\mathbf{o}, \mathbf{a}_\tau, \tau) - \mathbf{z}||_2^2 \right], \tag{1}$$

where the expectation is over all denoising timesteps, $\tau \sim \mathcal{U}[1, T]$, and observations and actions are drawn from a demonstration dataset, $\mathbf{o}, \mathbf{a} \sim \mathcal{D}$.

At sampling time, with further variance schedule parameters $\alpha_\tau$ & $\sigma$, inputs are iteratively denoised:

$$\mathbf{a}_{\tau-1} = \frac{1}{\sqrt{\alpha_\tau}} \left( \mathbf{a}_\tau - \frac{1 - \alpha_\tau}{\sqrt{1 - \bar{\alpha}_\tau}} \epsilon(\mathbf{o}, \mathbf{a}_\tau, \tau) \right) + \sigma_\tau \mathbf{z}. \tag{2}$$

### 3.2 ARCHITECTURAL DESIGN

This section explores neural network architectures for observation-to-action diffusion models. Specifically, the requirements of the network are: **Input:** Noisy action $\mathbf{a}_{\tau-1} \in \mathbb{R}^{|\mathbf{a}|}$, denoising timestep $\tau$, observation $\mathbf{o}$ (possibly with a history); **Output:** Predicted noise mask, $\hat{\mathbf{z}} \in \mathbb{R}^{|\mathbf{a}|}$.

Figure 2: Diffusion BC generates an action vector conditioned on an observation (which may be an image). By contrast, text-to-image diffusion models generate an image conditioned on a vector.

While U-Nets have become standard components of text-to-image diffusion models, their use only makes sense for large, spatial input and outputs, while we require generation of an action vector of modest dimensionality. Therefore, we now describe three architectures of varying complexity. Section 4 empirically assesses these three architectures, finding performance improvements in the order: Basic MLP < MLP Sieve < Transformer.

**Basic MLP.** This architecture directly concatenates all relevant inputs together, $[\mathbf{a}_{\tau-1}, \mathbf{o}, \tau]$. This input is fed into a multi-layer perceptron (MLP).

**MLP Sieve.** This uses three encoding networks to produce embeddings of the observation, denoising timestep, and action: $\mathbf{o}^e, \mathbf{t}^e, \mathbf{a}^e_{\tau-1} \in \mathbb{R}^{\text{embed dim}}$. These are concatenated together as input to a denoising network, $[\mathbf{o}^e, \mathbf{t}^e, \mathbf{a}^e_{\tau-1}]$. The denoising network is a fully-connected architecture, with residual skip connections, and with the raw denoising timestep $\tau$ and action $\mathbf{a}_{\tau-1}$ repeatedly concatenated after each hidden layer. To include a longer observation history, previous observations are passed through the same embedding network, and embeddings are concatenated together.

**Transformer.** This creates embeddings as for MLP Sieve. A multi-headed attention architecture (Vaswani et al., 2017) (as found in modern transformer encoder networks) is then used as the denoising network. At least three tokens are used as input, $\mathbf{o}^e, \mathbf{t}^e, \mathbf{a}^e_{\tau-1}$, and this can be extended to incorporate a longer history of observations (only the current $\mathbf{t}^e, \mathbf{a}^e_{\tau-1}$ are needed since the diffusion process is Markovian).

**Sampling rate.** The MLP Sieve and Transformer are carefully designed so the observation encoder is separate from the denoising network. At test time, this means only a single forward pass is required for the observation encoder, with multiple forward passes run through the lighter denoising network. This results in a manageable sampling time – in the experiment playing a video game from pixels (section 4.2), we were able to roll out our diffusion models at 8Hz on an average gaming GPU (NVIDIA GTX 1060 Mobile). Table 6 provides a detailed breakdown. Note that sampling time is a more severe issue in text-to-image diffusion models, where forward passes of the heavy U-Net architectures are required for all denoising timesteps.

### 3.3 WHY CLASSIFIER-FREE GUIDANCE FAILS

Classifier-Free Guidance (CFG) has become a core ingredient for text-to-image models, allowing one to trade-off image typicality with diversity (Ho and Salimans, 2021). In CFG, a neural network is trained as both a conditional and unconditional generative model. During sampling, by introducing a 'guidance weight' $w$, one places a higher weight ($w > 0$) on the prediction conditioned on some context (here, $\mathbf{o}$), and a negative weight on the unconditional prediction,

$$\hat{\mathbf{z}}_\tau = (1 + w)\epsilon_{\text{cond.}}(\mathbf{a}_{\tau-1}, \mathbf{o}, \tau) - w\epsilon_{\text{uncond.}}(\mathbf{a}_{\tau-1}, \tau). \tag{3}$$

One might anticipate that CFG would also be beneficial in the sequential setting, with larger $w$ producing trajectories of higher likelihood, but at the cost of the diversity. Surprisingly, we find that CFG can actually encourage *less* common trajectories, and degrade performance.

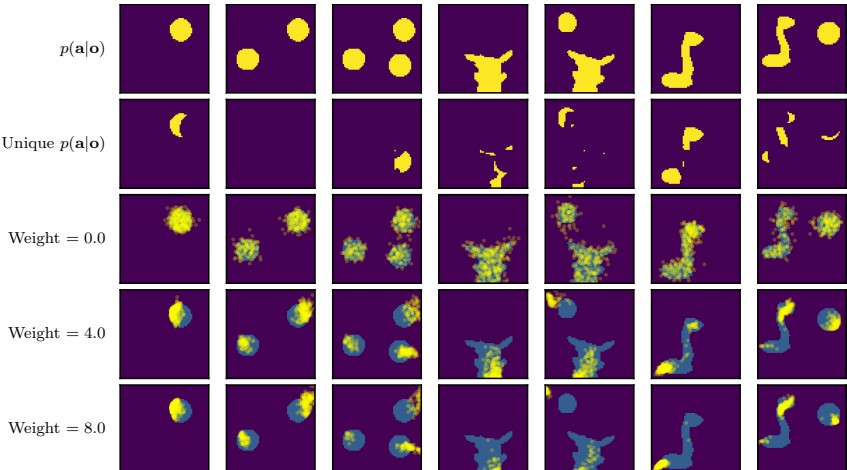

Figure 3: We vary the CFG 'weight' parameter ($w$ value in Eq. 3) during sampling in the Arcade Claw environment. CFG encourages selection of actions that were specific to an observation (maximising $p(\mathbf{o}|\mathbf{a})$). This can lead to less common trajectories being sampled more often.

In Figure 3 we visualise $\hat{p}(\mathbf{a}|\mathbf{o})$ for the claw machine game, under varying guidance strengths, $w$. An interpretation of CFG is that it encourages sampling of actions that would maximise an implicit classifier, $p(\mathbf{o}|\mathbf{a})$. Hence, CFG encourages selection of actions that were unique to a particular observation (Ho et al., 2022). Whilst this is useful for text-to-image models (generate images that are more specific to a prompt), in sequential environments this leads to an agent rejecting higher-likelihood actions in favour of less usual ones that were paired with some observation. In later experiments (section 4.1) we demonstrate empirically that this can lead to less common trajectories being favoured, while degrading overall performance. Appendix E provides a didactic example of when CFG fails in this way.

## 3.4 RELIABLE SAMPLING SCHEMES

In text-to-image diffusion, several samples are typically generated in parallel, allowing the user to select their favourite, and ignore any failures. However, when rolling out an observation-to-action diffusion model, such manual screening is not feasible. There remains a risk that a bad action could be selected during a roll-out, which may send an agent toward an out-of-distribution state. Hence, we propose 'Diffusion-X' and 'Diffusion-KDE' as variants of Diffusion BC, that mirror this screening process by encouraging higher-likelihood actions during sampling. For both methods, the training procedure is unchanged (only the conditional version of the model is required). The algorithms for all sampling methods are given in appendix D.

**Diffusion-X.** The sampling process runs as normal for $T$ denoising timesteps. The denoising timestep is then fixed, $\tau = 1$, and extra denoising iterations continue to run for $M$ timesteps. The intuition behind this is that samples continue to be moved toward higher-likelihood regions.

**Diffusion-KDE.** Generate multiple action samples from the diffusion model as usual (these can be done in parallel). Fit a simple kernel-density estimator (KDE) over all samples, and score the likelihood of each. Select the action with the highest likelihood.

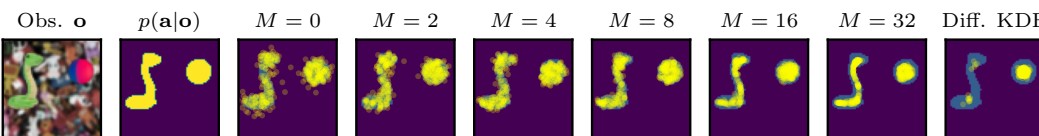

Figure 4: This figure shows the predictive distributions of diffusion models using various sampling schemes. When $M = 0$ this is 'Diffusion BC' and when $M > 0$ this is 'Diffusion-X'. Diff. KDE refers to 'Diffusion-KDE'.

The effect of these sampling modifcations is demonstrated in Figure 4. While Diffusion BC generates a small number of actions that fall outside of the true $p(\mathbf{a}|\mathbf{o})$ region, Diffusion-X and Diffusion-KDE avoid these bad actions. Note that the two distinct modes in the figure are recovered by both sampling methods, suggesting that multimodality is not compromised, though the diversity within each mode is reduced. Both techniques are simple to implement, and experiments in Section 4 show their benefit.

## 4 EXPERIMENTS

This section empirically investigates the efficacy of diffusion models for BC. We assess our method in two complex sequential environments, which have large human datasets available, with the aim of answering several questions: *Q1) How do diffusion models compare to existing baseline methods for BC, in terms of matching the demonstration distribution?* Section 4.1 compares to four popular modelling choices in BC, as well as recent state-of-the-art models. Section 4.2 provides further focused comparisons. *Q2) How is performance affected by the architectures designed in section 3.2, CFG, and the sampling schemes in section 3.4?* We provide full ablations in Section 4.1, and targeted ablations over sampling schemes in Section 4.2. *Q3) Can diffusion models scale to complex environments efficiently?* Section 4.2 tests on an environment where the observation is a high-resolution image, the action space is mixed continuous & discrete, and there are strict time constraints for sampling time.

**Evaluation.** To evaluate how closely methods imitate human demonstrations, we compare the behaviours of our models with those of the humans in the dataset. To do so, we will compare both at a high-level, analysing observable outcomes (e.g. tasks completions or game score), as well as at a low-level comparing the stationary distributions over states or actions. Both of these are important in evaluating how *humanlike* our models are, and provide complimentary analyses. Appendix B provides details on our evaluation metrics, which are introduced less formally here.

**Baselines.** Where applicable we include *Human* to indicate the metrics achieved by samples drawn from the demonstration dataset itself. We then re-implement five baselines. Three correspond to popular BC modelling choices, namely *MSE*: a model trained via mean squared error; *Discretised*: each action dimension is discretised into 20 uniform bins, then trained independently via cross-entropy; and *K-means*: a set of $K$ candidate actions are first produced by running K-means over all actions, actions are discretised into their closest bin, and the model is trained via cross-entropy. A further two baselines can be considered strong, more complex methods, namely *K-means+Residual*: as with K-means, but additionally learns a continuous residual on top of each bin prediction, trained via MSE, which was the core innovation of Behaviour Transformers (BeT) (Shafiullah et al., 2022); and *EBM*: a generative energy-based model trained with a contrastive loss, proposed in (Florence et al., 2022) – full details about the challenges of this method given in Appendix B.4.

One of our experiments uses the set up from Shafiullah et al. (2022), allowing us to compare to their reported results, including *Behaviour Transformers (BeT)*: the K-mean+residual combined with a large 6-layer transformer, and previous 10 observations as history; *Implicit BC*: the official implementation of energy-based models for BC (Florence et al., 2022); *SPiRL*: using a VAE, originally from Pertsch et al. (2021); and *PARROT*: a flow-based model, originally from Singh et al. (2020).

**Architectures & Hyperparameters.** We trial the three architecture options described in Section 3.2, which are identical across both the five re-implemented baseline methods and our diffusion variants (except where required for output dimensionality). Basic MLP and MLP Sieve both use networks with GELU activations and 3 hidden-layers of 512 units each. Transformer uses four standard encoder blocks, each with 16 attention heads. Embedding dimension is 128 for MLP Sieve and Transformer. Appendix B gives full hyperparameter details.

### 4.1 LEARNING ROBOTIC CONTROL FROM HUMAN DEMONSTRATION

In this environment, an agent controls a robotic arm inside a simulated kitchen. It is able to perform seven tasks of interest, such as opening a microwave, or turning on a stove (Appendix B contains full details). The demonstration dataset contains 566 trajectories. These were collected using a virtual reality setup, with a human's movements translated to robot joint actuations (Gupta et al.,

2020). Each demonstration trajectory performed four predetermined tasks. There are 25 different task sequences present in the dataset, of roughly equal proportion.

This environment has become a popular offline RL benchmark for learning reward-maximising policies (Fu et al., 2020). However, as our goal is to learn the full distribution of demonstrations, we instead follow the setup introduced by Shafiullah et al. (2022), which ignores any goal conditioning and aims to train an agent that can recover the full set of demonstrating policies.

The kitchen environment's observation space is a 30-dimensional continuous vector containing information about the positions of the objects and robot joints. The action space is a 9-dimensional continuous vector of joint actuations. All models receive the previous two observations as input, allowing the agent to infer velocities. For diffusion models, we set $T = 50$.

The kitchen environment is challenging for several reasons. 1) Strong (sometimes non-linear) correlations exist between action dimensions. 2) There is multimodality in $p(\mathbf{a}|\mathbf{o})$ at the point the agent selects which task to complete next, and also in how it completes it. We show that our diffusion model learns to represent both these properties in Figure 9, which visualises relationships between all action dimensions during one rollout.

### 4.1.1 MAIN RESULTS

Comparing the behaviour of our agents with that of the humans demonstrations is a challenging research problem in its own right. Table 1 presents several metrics that provide insight into how closely models match the human demonstrations – from high-level analysis of task sequences selected, to low-level statistics of the observation trajectories generated by models. We briefly summarise these, more technical descriptions can be found in Appendix B. Training and sampling times of methods are given in Table 6.

**Tasks$\geq$4.** We first measure the proportion of rollouts for which models perform four valid tasks, which nearly all human demonstrations achieve.

**Tasks Wasserstein.** For each method we record how many times each different task sequence was completed during rollouts. The Wasserstein distance is then computed between the resulting histogram, compared to the human distribution.

**Time Wasserstein.** As well as analysing which tasks were completed, we also analyse *when* they were completed. Figure 5 plots the distribution of the time taken to complete different numbers of tasks (normalised to exclude failures), for MSE and Diffusion-X. We then compute the Wasserstein distance between the human and agent distributions (see Appendix C for a full break down.)

**State Wasserstein.** If we truly capture the full diversity and fidelity of human behaviour present in the dataset, this will be reflected in the state occupancy distribution. We compute the Wasserstein between agent and human distributions, with a lower value indicating they are more similar.

**Density & Coverage.** We use the 'Density' and 'Coverage' metrics from Naeem et al. (2020) that are used to evaluate GANs. Roughly speaking 'Density' corresponds to how many states from *human* trajectories are close to the *agent's* states and 'Coverage' corresponds to the proportion of *human* states that have an *agent* generated state nearby.

In terms of task-completion rate, our Diffusion BC approaches outperform all baselines, with the ordering; Diffusion BC<Diffusion X<Diffusion-KDE. This ordering supports our hypothesis that these sampling schemes more reliably avoid bad actions. These improvements come at the cost of increased sampling time – for MLP Sieve, sampling rate drops from 16 Hz for Diffusion BC to 14 Hz for Diffusion-X to 12 Hz for Diffusion-KDE (Table 6).

For all Wasserstein metrics, diffusion models again outperform other baselines, and sampling schemes are usually ordered Diffusion KDE<Diffusion BC<Diffusion-X. This is important; while Diffusion-KDE completes more tasks, it signals a tendency to overfit to a smaller number of task sequences, generating less of the diversity from the demonstration distribution. This is quantitatively confirmed by Diffusion-KDE scoring significantly higher Density, but lower Coverage.

**Architecture ablation.** In terms of the different architectures designed in Section 3.2, Table 1 shows that metrics usually improve in order: Basic MLP<MLP Sieve<Transformer. These improvements

Table 1: Robotic control results. Mean $\pm$ one standard error over three training runs (100 rollouts). Methods marked with asterisk (*) are our proposed methods.

| | Tasks $\geq 4 \uparrow$ | Tasks Wasserstein $\downarrow$ | Time Wasserstein $\downarrow$ | State Wasserstein $\downarrow$ | Density $\uparrow$ | Coverage $\uparrow$ |
|---|---|---|---|---|---|---|
| **MLP Basic Architecture** | | | | | | |
| *Diffusion BC, Basic MLP | $0.45 \pm 0.03$ | $1.96 \pm 0.12$ | $12.04 \pm 2.20$ | $0.463 \pm 0.012$ | $0.54 \pm 0.02$ | $0.38 \pm 0.01$ |
| *Diffusion-KDE, Basic MLP | $0.59 \pm 0.01$ | $1.72 \pm 0.03$ | $8.08 \pm 0.24$ | $0.481 \pm 0.005$ | $0.78 \pm 0.00$ | $0.37 \pm 0.00$ |
| *Diffusion-X, Basic MLP | $0.58 \pm 0.02$ | $1.51 \pm 0.14$ | $8.61 \pm 0.14$ | $0.424 \pm 0.017$ | $0.64 \pm 0.00$ | $0.41 \pm 0.00$ |
| | | | | | | |
| **MLP Sieve Architecture** | | | | | | |
| MSE, MLP Sieve | $0.5 \pm 0.02$ | $1.91 \pm 0.07$ | $6.40 \pm 0.48$ | $0.443 \pm 0.021$ | $0.71 \pm 0.01$ | $0.40 \pm 0.01$ |
| Discretised, MLP Sieve | $0.18 \pm 0.02$ | $3.43 \pm 0.14$ | $11.30 \pm 1.29$ | $0.651 \pm 0.026$ | $0.38 \pm 0.02$ | $0.31 \pm 0.01$ |
| K-Means, MLP Sieve | $0.0 \pm 0.0$ | $5.25 \pm 0.0$ | – | $1.469 \pm 0.120$ | $0.09 \pm 0.00$ | $0.06 \pm 0.00$ |
| K-Means+Residual, MLP Sieve | $0.23 \pm 0.02$ | $2.87 \pm 0.16$ | $11.60 \pm 2.11$ | $0.607 \pm 0.027$ | $0.51 \pm 0.01$ | $0.36 \pm 0.00$ |
| EBM Deriv-Free, MLP Sieve | $0.0$ | – | – | – | – | – |
| *Diffusion BC, MLP Sieve | $0.68 \pm 0.02$ | $1.31 \pm 0.05$ | $6.06 \pm 1.10$ | $0.373 \pm 0.012$ | $0.66 \pm 0.01$ | $0.42 \pm 0.00$ |
| *Diffusion-KDE, MLP Sieve | $0.79 \pm 0.04$ | $1.6 \pm 0.24$ | $6.77 \pm 0.64$ | $0.439 \pm 0.039$ | $0.93 \pm 0.02$ | $0.41 \pm 0.01$ |
| *Diffusion-X, MLP Sieve | $0.77 \pm 0.02$ | $\mathbf{1.06 \pm 0.05}$ | $5.24 \pm 0.90$ | $0.344 \pm 0.004$ | $0.78 \pm 0.01$ | $\mathbf{0.45 \pm 0.00}$ |
| | | | | | | |
| **Transformer Architecture** | | | | | | |
| MSE, Transformer | $0.69 \pm 0.02$ | $1.47 \pm 0.13$ | $5.85 \pm 0.27$ | $0.397 \pm 0.034$ | $0.81 \pm 0.01$ | $0.42 \pm 0.01$ |
| Discretised, Transformer | $0.34 \pm 0.02$ | $2.54 \pm 0.14$ | $6.13 \pm 0.49$ | $0.512 \pm 0.002$ | $0.47 \pm 0.01$ | $0.36 \pm 0.00$ |
| K-Means, Transformer | $0.0$ | $5.25$ | – | $1.470$ | $0.07$ | $0.06$ |
| K-Means+Residual, Transformer | $0.34 \pm 0.02$ | $2.25 \pm 0.16$ | $7.80 \pm 0.87$ | $0.426 \pm 0.018$ | $0.66 \pm 0.02$ | $0.38 \pm 0.01$ |
| *Diffusion BC, Transformer | $0.77 \pm 0.01$ | $1.35 \pm 0.11$ | $\mathbf{4.11 \pm 0.05}$ | $\mathbf{0.340 \pm 0.003}$ | $0.74 \pm 0.01$ | $0.44 \pm 0.00$ |
| *Diffusion-KDE, Transformer | $\mathbf{0.89 \pm 0.01}$ | $1.31 \pm 0.03$ | $5.28 \pm 0.41$ | $0.418 \pm 0.012$ | $\mathbf{0.97 \pm 0.02}$ | $0.43 \pm 0.01$ |
| *Diffusion-X, Transformer | $0.88 \pm 0.01$ | $1.17 \pm 0.13$ | $4.65 \pm 0.47$ | $0.365 \pm 0.013$ | $0.94 \pm 0.02$ | $\mathbf{0.45 \pm 0.01}$ |
| | | | | | | |
| **From Shafiullah et al. (2022)** | | | | | | |
| Behaviour Transformers | $0.44$ | | | | | |
| Implicit BC | $0.24$ | | | | | |
| SPiRL VAE | $0.0$ | | | | | |
| PARROT Normalizing Flow | $0.0$ | | | | | |
| | | | | | | |
| **Dataset** | | | | | | |
| Human | $0.98$ | | | | | |
| Human sub-sampled | – | – | – | $0.223 \pm 0.006$ | $1.00 \pm 0.01$ | $0.56 \pm 0.01$ |

do come at the cost of increased training and inference time – for Diffusion BC the sampling rate drops from 24 Hz for MLP Basic to 16 Hz for MLP Sieve to 4 Hz for transformer (Table 6).

### 4.1.2 CLASSIFIER FREE GUIDANCE ANALYSIS

Section 3.3 provided intuition for why we expect CFG to fail in observation-to-sequence models. We now test our hypotheses empirically. During training the observation embedding is randomly masked with zeros with probability 0.1. This allows us to learn both a conditional and unconditional generative model. We then perform rollouts under different guidance weights, $w \in \{0, 1, 4, 8\}$, measuring the rate of completion of 4 tasks and monitoring which task the agent completes first.

Table 2 shows that completion rate drops from 0.63 without guidance ($w = 0$) to 0.08 with strong guidance ($w = 8$). Meanwhile, CFG also creates a strong bias towards selection of Bottom Burner as the first task. This is significant as the human demonstrators select Bottom Burner just 10% of the time, but this increases from 7% (no guidance, $w = 0$) to 25% (strong guidance, $w = 8$), showing that CFG encourages less usual trajectories.

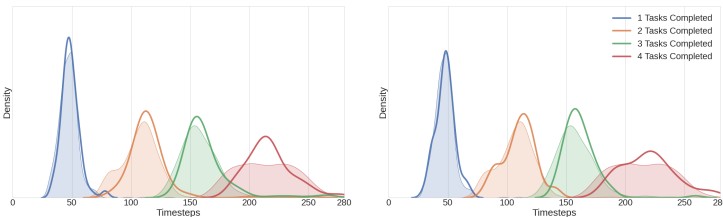

Figure 5: Time to complete robotic control kitchen tasks. **Shaded:** Human demonstrations. **Left:** MSE Transformer, **Right:** Diffusion-X Transformer.

Table 2: Effect of CFG for Diffusion BC, MLP Sieve. Mean over three training runs (100 rollouts).

| | | First task peformed | | | |
|---|---|---|---|---|---|
| | Tasks $\geq 4\uparrow$ | Microwave | Kettle | Bottom Burner | Other/Failure |
| Guidance $w = 0.0$ | **0.63** | 62.3 | 29.7 | 7.3 | 0.7 |
| Guidance $w = 1.0$ | 0.61 | 57.3 | 30.0 | 12.7 | 0.0 |
| Guidance $w = 4.0$ | 0.45 | 52.7 | 28.3 | 17.0 | 2.0 |
| Guidance $w = 8.0$ | 0.08 | 24.0 | 26.7 | 24.7 | 25.6 |
| Human Demonstrations | **0.63** | 58.3 | 31.6 | 10.1 | 0.0 |

## 4.2 MODELLING HUMAN GAMEPLAY IN A VIDEO GAME

We further tested our models in the environment 'Counter-Strike: Global Offensive' (CSGO) introduced by Pearce and Zhu (2022) (`https://github.com/TeaPearce/Counter-Strike_Behavioural_Cloning`). CSGO is one of the world's most popular video games in player and spectator numbers. We use the 'aim train' environment, where a player is fixed on a platform in the center of a map, and must defend themselves against enemies controlled by built-in AI, who rush towards the player. Success requires precise and coordinated control over a mixed continuous & discrete action space, as well as dealing with multimodality in target selection and aiming.

The demonstration dataset contains 45,000 observation/action tuples, recorded from a high-skill human player. Observations are $280\times150$ RGB images, and the action space is three dimensional – mouse $x \in \mathbb{R}$, mouse $y \in \mathbb{R}$, left click $\in \{0, 1\}$. The environment runs asynchronously at a fixed rate, providing a rigorous test of sampling speed of models. Due to this constraint, we test only the MLP Sieve architecture, which offers a good trade-off between inference speed and performance. We also exclude the slightly slower Diffusion-KDE sampling method. $T$ is set to 20.

For baselines, we compare to MSE, discretised, and K-Means+Residual which were the strongest baselines in the kitchen environment. We also include EBM using derivative-free optimisation. We consider two options for the observation encoder: 1) a lightweight CNN, and 2) a ResNet18 with ImageNet weights. Diffusion models use 20 denoising timesteps, and Discrete uses 20 bins per action dimension. For MSE, we optimise the left click action via binary cross-entropy and mouse $x$ & $y$ via MSE. Appendix B provides further details on hyperparameters and metrics.

Table 3 reports results. Diffusion-X performs best across both observation encoders in terms of game score and distance to the human distribution, which was measured as the Wasserstein distance on the actions predicted (grouped into sequences of different lengths to assess the temporal consistency). Training and sampling times of methods are given in Table 6. Diffusion-X runs at 18 Hz compared to 200 Hz for MSE. Training time is similar.

Table 3: Video game results, mean over three rollouts of 10 minutes. Methods marked with asterisk (*) are our proposed methods.

| | | Wasserstein Distance, Human to Model $\downarrow$ | | |
|---|---|---|---|---|
| | Game Score $\uparrow$ | 1$\times$timesteps | 16$\times$timesteps (1 sec) | 32$\times$timesteps (2 sec) |
| **Observation encoder:** 6-layer CNN | | | | |
| MSE, MLP Sieve | 6.9 | 19.0 | 37.6 | 54.7 |
| Discrete, MLP Sieve | 7.8 | 12.9 | 35.0 | 55.9 |
| *Diffusion BC, MLP Sieve | 12.9 | 14.3 | 34.5 | 52.8 |
| *Diffusion-X, MLP Sieve | **17.3** | **11.7** | **30.0** | **47.8** |
| **Observation encoder:** ResNet18 | | | | |
| MSE, MLP Sieve | 17.8 | 5.5 | 28.1 | 48.9 |
| Discrete, MLP Sieve | 14.7 | 6.6 | 31.3 | 53.0 |
| K-Means+Residual, MLP Sieve | 16.8 | **3.8** | 29.2 | 51.8 |
| EBM Derivative-Free, MLP Sieve | 4.3 | 17.2 | 50.0 | 74.1 |
| *Diffusion BC, MLP Sieve | 19.0 | 6.3 | 29.5 | 50.4 |
| *Diffusion-X, MLP Sieve | **24.0** | 4.5 | **24.5** | **44.4** |
| **Baselines** | | | | |
| Human | 36.5 | 0.73 | 0.57 | 0.38 |

## 5 DISCUSSION & CONCLUSION

This paper provided the first thorough investigation into using diffusion models for imitating human behaviour. Existing modelling choices in BC make various approximations that simplify the learning problem. But these approximations come at a cost: they introduce limitations and biases on the cloned policy, which we systematically identified in section 2. Recent developments in generative modelling for images have shown that diffusion models are capable of learning arbitrarily complex conditional distributions, and are stable to train. This paper has combined these two worlds, proposing that diffusion models are also an excellent fit for learning the complex observation-to-action distributions found in datasets of humans behaving in sequential environments. Our key insight is that, unlike many modelling choices for BC, diffusion models make no coarse approximations on the target distribution, so avoid many of their shortcomings.

Diffusion models do come with several limitations. They increase inference time – MSE model can sample actions at 666 Hz and 200 Hz in the kitchen and CSGO environments, while Diffusion BC samples at 16 Hz and 32 Hz. They also introduce several new hyperparameters. Finally, our diffusion models address only one challenge in imitation learning – that of learning complex action distributions at a single environment timestep. There are other open challenges they do not address such as learning correlations *across* environment timesteps.

This paper contributed several innovations to successfully adapt diffusion models to the observation-to-action domain. Our experiments demonstrated the effectiveness of these with several key takeaways. 1) Diffusion models offer improvements over other methods in matching the demonstrations in terms of reward and distribution. 2) Reliable sampling schemes Diffusion-X and Diffusion-KDE offer benefits over Diffusion BC. 3) Good architecture design is important to the success of Diffusion models, allowing trade-off of performance and sampling speed. 4) CFG should be avoided as a mechanism to condition on observations for diffusion agents in sequential environments.

Experiments support our hypothesis that, by avoiding coarse approximations about the action distribution, diffusion models improve over existing methods at modelling human demonstrations. On a complex control task, diffusion models achieved a task completion rate of 89%, exceeding recent state-of-the-art of 44%. We also demonstrated that our models can scale to learn directly from image observations, on a mix of continuous and discrete actions, sampling actions at 18 Hz. This demonstrates the possibility of applying our models in complex, real-world settings.

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

This appendix is organised as follows.

## A  EXTENDED RELATED WORK

**Implicit BC.** Florence et al. (2022) formulated BC as a conditional energy-based modeling problem. They showed that the ability of energy-based policies to represent multimodal distributions and discontinuous functions can provide competitive results with state-of-the-art offline reinforcement learning methods. Our paper is complimentary to this work, continuing the investigation on expressive models, studying the fit of diffusion models for BC. As well as showing that they can outperform state-of-the-art baselines in terms of score, we also show they outperform in closeness to the demonstration distribution, which was not explored by Florence et al. (2022). We note that diffusion models avoid some of the challenges of energy-based models – they remove the requirement to generate negative samples during training, and avoid some of the complexities in the sampling process that energy-based models face.

**Transformer + BC.** Although there are multiple works using transformers for RL, in this section we only consider approaches that use transformers for BC, where no interaction with the environment is allowed, and only state-action trajectories and no reward signal are available in a dataset, for which the existing literature is scarce.

Reed et al. (2022) proposed GATO, an agent that could model different input modalities, including text, images and behaviour policies. However, they focused on demonstrating multitask capabilities, rather than on on how to do BC efficiently. GATO relies on discretisation, which can lead to inaccurate actions due to quantisation errors (see Sec. 2); and on autoregressive predictions, which can lead to repetitive and out of distribution trajectories (Holtzman et al., 2019).

Shafiullah et al. (2022) proposed a transformer based BC method that implements the K-Means+Residual approach discussed in Sec. 2. This is done by using two heads, one with a categorical loss for predicting the centroid of the cluster to which the action belongs; and the other one with an MSE loss to predict the 'residual', that is the distance from the centroid to the action. This resulted in a discrete approximation of the policy. Although having limitations, it was able to outperform other approximations in a number of environment, using both human and synthetic data.

Here, we focus on generative modelling with diffusion models, evaluating the impact of different neural network architectures and different sampling schemes, and showing they are expressive enough to model the policy without implicit approximations, outperforming previous approaches, including discretisation and K-Means+Residual.

**Diffusion models.** One main difference with previous works on diffusion models is that in order to obtain high quality samples, they relied on guiding the sampling process, either by using an expensive classifier (Dhariwal and Nichol, 2021), or by training a conditional and an unconditional model together (Ho and Salimans, 2021). However, as explained in Sec. 3.2, guidance can bias sampling towards low likelihood trajectories, which is problematic for BC applications, so in order to obtain high quality and high likelihood samples, we introduce two novel guidance-free methods.

**RL and Diffusion Models.** Up to our knowledge, the only previous work that used diffusion models for RL is by Janner et al. (2022) who proposed Diffuser: a model that predicts all states and actions of a complete trajectory; and uses classifier-like guidance for trajectory optimization and similar U-Net architectures to those from text-to-image diffusion models (Ramesh et al., 2022; Saharia et al., 2022), replacing the spatial convolutions with temporal convolutions. The main limitation of Diffuser is that sampling complete trajectories is computationally expensive, especially since the trajectory has to be re-sampled at each timestep (open-loop). Hence, it was demonstrated for environments with low-dimensional state-action sets. Our model is able to generate a trajectory by

sampling one step at a time (closed loop). Combined with replacing U-Nets with other architectures like MLPs and transformer, this enables learning from high-dimensional states like images.

Concurrent to our work, Wang et al. (2022) use a conditional diffusion model with an MLP to predict a single action at a time conditioned on the current state. However, their focus is on trajectory optimization, relying on bootstrapping to learn a value function that is used for (classifier-like) guidance, and test their approach in environments with low dimensional state sets. Here, we focus on BC, for which we observe guidance can bias the sampling towards low-likelihood trajectories, so we introduce two novel guidance-free conditional sampling methods. In addition, we explore the impact of multiple architectures, including transformers, and demonstrate their performance even under high-dimensional state sets.

Ajay et al. (2023) provide a further concurrent work. They propose to diffuse sequences of future states, using a separate inverse-dynamics model to then infer actions. This contrasts with our own approach of directly diffusion actions, which allows us to scale up to image observations. They also explore conditioning on reward, skills and constraints, finding that CFG can help enforce the conditioning. By contrast, our investigation considered using CFG when the *observation* is the conditioning variable.

**Imitation Learning (IL).** This paper lies under the IL paradigm (Hussein et al., 2017), in particular the line of work that frames IL as matching the state-action occupancy distribution in the dataset, which is related to Inverse RL (Ho and Ermon, 2016; Ghasemipour et al., 2020; Ke et al., 2020). Indeed, we use diffusion models as a practical BC approach that optimise a variational bound of the usual BC objective (which can be interpreted as minimising the forward KL w.r.t. to the experts distribution Ghasemipour et al. (2020); Ke et al. (2020)), but, as discussed in Sec. 2, diffusion models are able to approximate multimodal distributions, surmounting one major limitation of previous BC methods (Ke et al., 2020), especially when working with real world datasets from multiple demonstrators.

Finally, similarly to Dadashi et al. (2020); Hussenot et al. (2021), we compute the Wasserstein distance between the occupancy distributions as an evaluation criteria.

# B   EXPERIMENTAL DETAILS

## B.1   CLAW ENVIRONMENT

The claw environment consists of seven unique pictures, each with different combination of valid toys to pick. To create the dataset, we pick one of the seven images at random, and then pick a demonstration action from the true $p(\mathbf{a}|\mathbf{o})$. The final dataset consists of 20000 such demonstrations.

Figure 8 shows full comparison of all methods in the seven different images. Each individual point represents one sample from the trained model when it is given the image observation $\mathbf{o}$.

We train all methods for 100 epochs with a batch size of 32, using Adam optimizer and a decaying learning rate starting from 0.0001. We use 50 diffusion steps for the diffusion models. We set $K = 10$ for K-Means clustering.

## B.2   KITCHEN

The seven tasks of interest are: open the microwave ('microwave'), move kettle onto a stove ('kettle'), turn on the bottom stove ('bottom burner'), turn on the top stove ('top burner'), turn on a light-switch ('light-switch'), open a sliding cabinet ('slide cabinet'), open a swinging door ('hinge cabinet').

For each method we train 3 different seeds, and roll out 100 trajectories of length 280 for evaluation. K-Means Transformer only has a single seed.

**Time Wasserstein:** For each method the histogram over the number of timesteps taken to complete N tasks is computed. For each tasks, the Wasserstein distance between the the histogram of the model's timesteps and the humans is computed. We then average those Wassersteins across all 4 tasks. In the situation where a method did not ever complete 4 tasks, we leave the entry blank.

**State-based Wasserstein:** We are approximating the Wasserstein between the stationary distributions over the state space of the human dataset and our models: $\mathcal{W}(\rho_{\text{AGENT}}(s), \rho_{\text{HUMAN}}(s))$. To do so we produce a 'dataset' for each method by concatenating the states for all 100 trajectories (similarly for the humans). We then turn these into the empirical distributions by having a uniform mass over each data point. We then use the POT library's `emd2` function (Flamary et al., 2021) to compute the Wasserstein distance, using the $L_2$ cost function between the 30 dimensional entries.

To produce the sub-sampled human dataset, we pick 100 trajectories uniformly without replacement from the human dataset. We do this 5 times.

**Density and Coverage:** We use the Density and Coverage metrics proposed in Naeem et al. (2020), and compute them using the code provided by the authors. We use the human dataset as the set of *real* state samples, and our model's rollouts as the *fake* state samples. We use 10 nearest neighbours for the calculation.

**Hyperparameters.** Neural network architectures were identical across all methods (except for the final linear layer which allows for differing output dimensionality). Basic MLP and MLP Sieve used three hidden-layers of 512 nodes and GELU activations. Embedding dimension of observation, action, and timestep was fixed at 128. Embedding networks were single hidden-layer MLPs with 128 nodes, using leaky ReLU activations except for the timestep encoder which used Sinusoidal activations. The Transformer used four encoding blocks, each with 16 self-attention heads and an internal embedding dimension of 64. All models receive the previous two observations as history.

Other hyperparameters were kept consistent across methods as far as possible. Initial experimentation showed that methods did not suffer from overfitting, and as default we trained models for 500 epochs with a cosine learning rate decay. MLP models used a learning rate of 1e-3 and batchsize of 512, while transformer models used a learning rate of 5e-4 and batchsize of 1024. We set $K = 64$ for K-means and discretised used 20 bins per action dimension.

We made extensive efforts to bring performance of K-means+residual inline with that reported in Shafiullah et al. (2022). Our best settings used a fixed learning rate of 1e-4, with $K = 64$, though this still gave a task completion rate of 0.34, which falls slightly below 0.44 as originally reported. The difference might be explained by the larger network used by Shafiullah et al. (2022) – 6 layers, and an observation history of 10 steps. Note we also tested hyperparameters as reported in their paper (50 epochs, batchsize of 64), but this provided worse performance for us.

For diffusion models, we set $T = 50$ and standard $\beta$ schedules linearly decaying in [1e-4, 0.02]. (Note that the variance schedule parameters $\alpha_t$, $\bar{\alpha}_t$ and $\sigma_t$ introduced in Section 3 are derived from this $\beta$ schedule.) For Diffusion-X, $M = 8$. For Diffusion-KDE, we first sample 100 actions, and fit a Gaussian KDE model (width=0.4) from scikit-learn[1].

Models were rolled out with 100 random seeds. Each episode lasted 280 timesteps as in Shafiullah et al. (2022) – 98% of humans completed their assigned four tasks within this time.

## B.3 CSGO

**Game Score:** Frags-per-minute, as reported by Pearce and Zhu (2022).

**Action-based Wasserstein:** Similarly to the Kitchen environment above, we are approximating the stationary distribution over the *action* space of the human dataset and our models: $\mathcal{W}(\rho_{\text{AGENT}}(a), \rho_{\text{HUMAN}}(a))$.

For `1xtimesteps`, we compute this in the same way as the state-based Wasserstein above, using POT's `emd2` and the $L_2$ cost between the 3 dimensional entries.

For `16xtimesteps` and `32xtimesteps`, for each timestep we concatenate the 16 (and 32 timesteps respectively) following it into a single vector. If there are not enough timesteps left in the trajectory, then we do not use that timestep. We compute the Wasserstein in the exact same manner as before on these higher dimensional entries.

**Hyperparameters.** Neural network architectures were unchanged from the Kitchen experiments, except for the observation encoder, for which we tested two options. 1) A lightweight 6-layer CNN

---

[1] https://scikit-learn.org/stable/

that operates directly on the frames stacked together. 2) A ResNet18 with ImageNet weights – the four stacked frames are passed through this independently. The output is then concatenated together and passed through two vanilla convolutional layers. Following average pooling, an embedding vector of length 128 is output.

Unlike in the Kitchen environment, we found models were sensitive to overfitting in CSGO. For the smaller observation encoder, we trained for 500 epochs with cosine decay (learning rate of 1e-4), but tested models both midway through training (250 epochs) and at the end (500 epochs). For the ResNet encoder, we trained with a fixed learning rate of 1e-4 for a maximum of 100 epochs, but tested models every 20 epochs. All methods were best with either 60 or 80 training epochs.

For diffusion models, we set $T = 20$ and standard $\beta$ schedules linearly decaying in [1e-4, 0.02]. For Diffusion-X, $M = 16$. For K-Means+Residual, we use $K = 64$.

Following Pearce and Zhu (2022), episodes were defined as 10 minutes of playing time. The original data was collected at 16Hz with the game running in real-time. We altered the game setting so that it runs at half speed, allowing our models to roll out at 8Hz. It was possible to increase this to 10Hz for diffusion BC.

## B.4 ENERGY-BASED MODEL IMPLEMENTATION

We re-implemented two versions of the Energy-Based Model's (EBM's) described in Implicit BC (Florence et al., 2022): 1) Derivative-free optimisation ('Deriv-free') and 2) Langevin MCMC ('Langevin'). Whilst we made efforts to run both of these in all three of our environments (Claw, Kitchen, and CSGO), we were unable to produce rudimentary results in many combinations but document our experiences here. Ultimately we reported EBM Deriv-free in Claw (Figure 8), Kitchen (Table 1 & 4) and CSGO (Table 3), and EBM Langevin on the Claw only (Figure 8). Other combinations failed to produce any reward.

**Derivative-Free Optimisation.** This follows Algorithm 1 of (Florence et al., 2022). At training time, negative examples are sampled uniformly from the action space, and a contrastive loss is optimised. At sampling time, a derivative-free optimisation scheme is used (reminiscent of particle-swarm optimisation).

We used the following hyperparameters by default: number counter examples=256, number inference samples=16000, number optimising iterations=3, optimising noise=0.33, noise decay=0.5.

In our experience this method performed reasonably in lower dimensional environments. In the Claw (Figure 8), we could recover a fair approximation of the true $p(\mathbf{a}|\mathbf{o})$ distributions, although there is a segmentation effect. To understand why this occurs, we plotted the energy mesh over the whole action space in Figure 6, where we can observe halo-like effects at the edges of objects, explaining this.

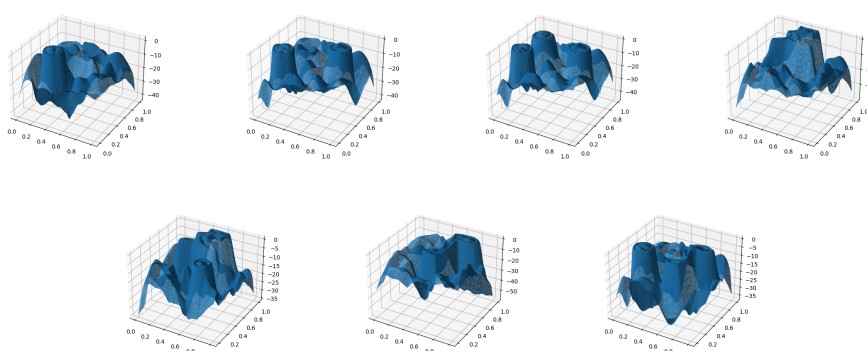

Figure 6: Visualisation of the energy landscapes learnt on the Claw environment by EBM derivative-free optimisation.

Derivative-free worked poorly in the kitchen environment which has a 9 dimensional action space, only occasionally completing two tasks and never more. This was also observed in Florence et al. (2022), which showed that Derivative-free failed when action dimensionality exceeded six.

Since CSGO's action space is of dimension three, we thought Derivative-free may perform reasonably, though we were only able to recover a rudimentary performance level as in Table 3.

Sampling times of Derivative-free were reasonable (Table 6), generally falling between diffusion models and MSE, though it slowed down training significantly since it requires drawing many negative samples for each datapoint in a batch – this also sometimes reduced the maximum batchsize that could be used (since it effectively gets multiplied).

**Langevin MCMC.** This follows the description in Appendix B.3 of Florence et al. (2022), along with the gradient penalty in B.3.1. Negative samples are partially optimised at training time, by backpropagating gradients to the action samples. At sampling time, the same gradient-based optimisation procedure is run for longer. Florence et al. (2022) state that this method should handle higher-dimensional action spaces well.

We used the following hyperparameters by default: number counter examples=32, noise scale = 0.5, learning coefficient start = 0.05, learning coefficient end = 0.005, number mcmc iterations during training=100, extra mcmc iterations during sampling=100, M=1.

We were unable to get this method working well in any of our environments. Figure 7 shows the energy landscapes learnt in the Claw. We found it produced energy functions that were overly smooth. Also, at sampling time, many samples would get stuck in local maxima at the edges of the action space. A fundamental issue with the approach is that the negative samples become less 'negative' over the course training as the model learns the energy function. This creates distribution drift in the training data and a less stable optimisation objective.

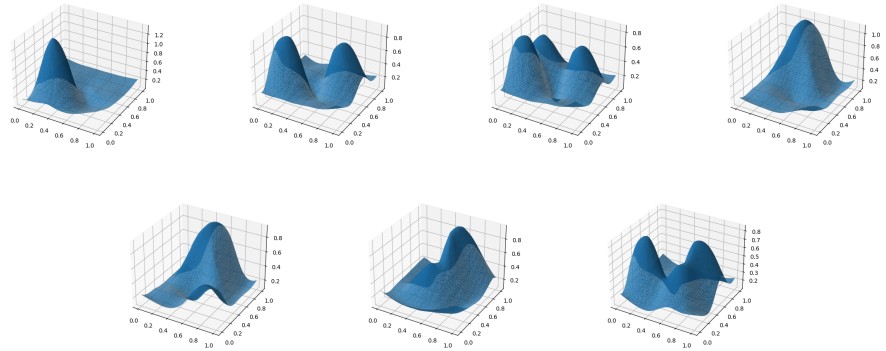

Figure 7: Visualisation of the energy landscapes learnt on the Claw environment by EBM Langevin MCMC optimisation.

In initial experiments, Langevin failed to solve any tasks in the Kitchen or CSGO. Both training and sampling from Langevin are far slower than any other method (Table 6) – training requires running an MCMC chain for many iterations for negative samples (set as default to 100). Sampling requires double this number of steps, and additionally cannot be performed within `torch.nograd()`, since the gradients are required for optimisation.

## C FURTHER RESULTS

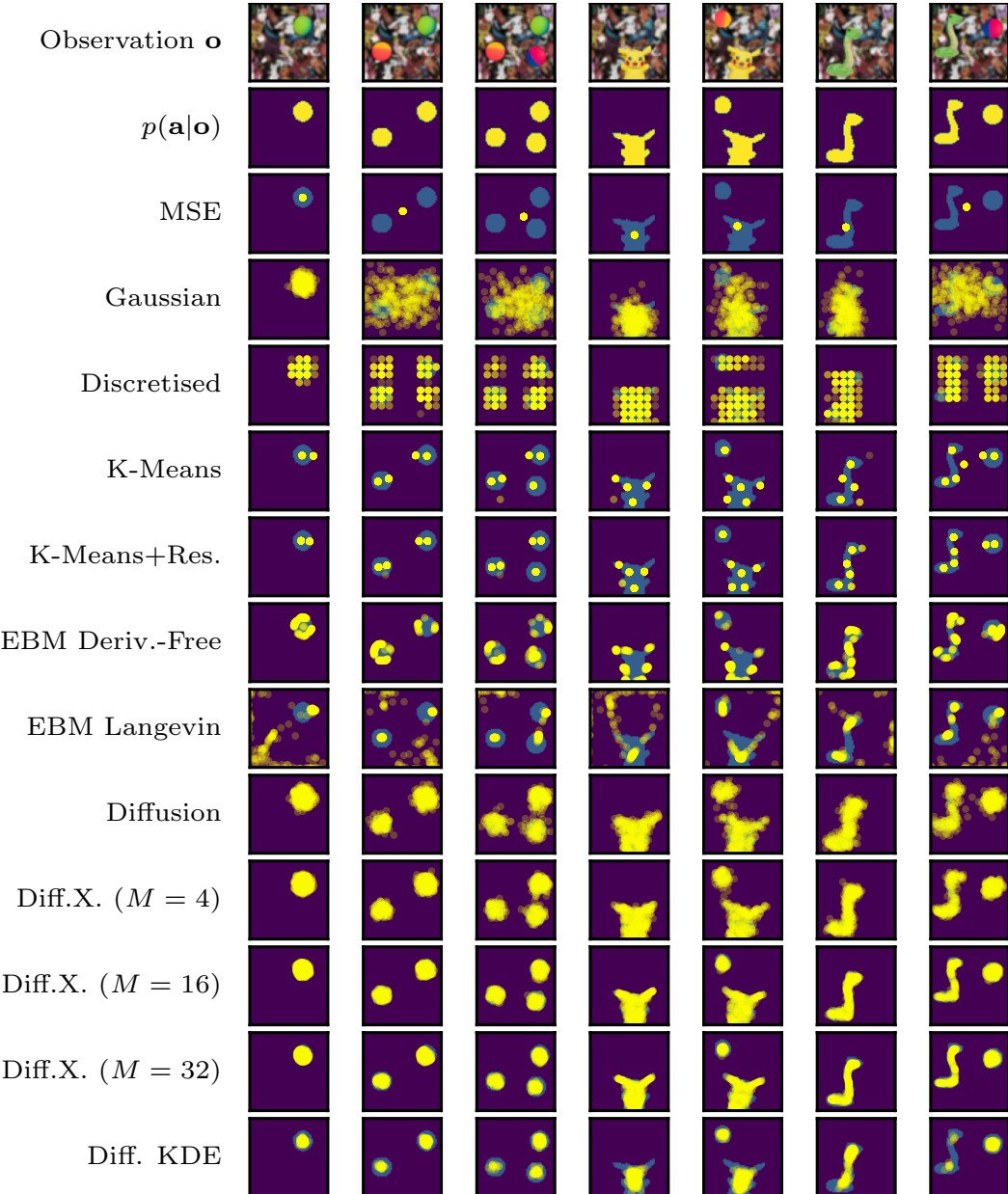

Figure 8: Full comparison of different distribution modelling choices in the toy claw environment.

Table 4: Robotic control results. Mean $\pm$ one standard error over three training runs (100 rollouts). These metric capture task completion only. Methods marked with asterisk (*) are our proposed methods.

| | No. runs | Tasks completed ↑ | Tasks $\geq 1$ ↑ | Tasks $\geq 2$ ↑ | Tasks $\geq 3$ ↑ | Tasks $\geq 4$ ↑ | Tasks $\geq 5$ ↑ |
|---|---|---|---|---|---|---|---|
| *Diffusion Basic MLP | 3 | $3.04 \pm 0.01$ | $0.98 \pm 0.01$ | $0.89 \pm 0.01$ | $0.71 \pm 0.0$ | $0.45 \pm 0.03$ | $0.01 \pm 0.0$ |
| *Diffusion Basic MLP KDE | 3 | $3.44 \pm 0.01$ | $0.99 \pm 0.0$ | $0.97 \pm 0.01$ | $0.86 \pm 0.0$ | $0.59 \pm 0.01$ | $0.04 \pm 0.01$ |
| *Diffusion Basic MLP Extra8 | 3 | $3.37 \pm 0.03$ | $0.99 \pm 0.01$ | $0.93 \pm 0.01$ | $0.83 \pm 0.02$ | $0.58 \pm 0.02$ | $0.04 \pm 0.01$ |
| | | | | | | | |
| MSE MLP Sieve | 3 | $3.13 \pm 0.07$ | $0.99 \pm 0.0$ | $0.9 \pm 0.03$ | $0.72 \pm 0.03$ | $0.5 \pm 0.02$ | $0.02 \pm 0.0$ |
| Discretised MLP Sieve | 3 | $2.26 \pm 0.08$ | $0.95 \pm 0.02$ | $0.73 \pm 0.02$ | $0.4 \pm 0.03$ | $0.18 \pm 0.02$ | $0.0 \pm 0.0$ |
| Disc. K-Means MLP Sieve | 3 | $0.33 \pm 0.03$ | $0.3 \pm 0.03$ | $0.03 \pm 0.0$ | $0.0 \pm 0.0$ | $0.0 \pm 0.0$ | $0.0 \pm 0.0$ |
| K-Means+Residual MLP Sieve | 3 | $2.56 \pm 0.04$ | $0.95 \pm 0.0$ | $0.82 \pm 0.01$ | $0.55 \pm 0.02$ | $0.23 \pm 0.02$ | $0.01 \pm 0.0$ |
| EBM Deriv-Free, MLP Sieve | 1 | $0.39$ | $0.31$ | $0.08$ | $0.0$ | $0.0$ | $0.0$ |
| *Diffusion MLP Sieve | 3 | $3.46 \pm 0.04$ | $0.99 \pm 0.0$ | $0.94 \pm 0.0$ | $0.82 \pm 0.02$ | $0.68 \pm 0.02$ | $0.02 \pm 0.01$ |
| *Diffusion KDE MLP Sieve | 3 | $3.78 \pm 0.06$ | $\mathbf{1.0 \pm 0.0}$ | $\mathbf{1.0 \pm 0.0}$ | $0.92 \pm 0.01$ | $0.79 \pm 0.04$ | $0.07 \pm 0.01$ |
| *Diffusion Extra MLP Sieve | 3 | $3.72 \pm 0.02$ | $\mathbf{1.0 \pm 0.0}$ | $0.97 \pm 0.0$ | $0.9 \pm 0.01$ | $0.77 \pm 0.02$ | $0.08 \pm 0.0$ |
| | | | | | | | |
| MSE Transformer | 3 | $3.59 \pm 0.02$ | $0.99 \pm 0.0$ | $0.96 \pm 0.01$ | $0.88 \pm 0.01$ | $0.69 \pm 0.01$ | $0.07 \pm 0.01$ |
| Discretised Transformer | 3 | $2.71 \pm 0.08$ | $0.97 \pm 0.0$ | $0.82 \pm 0.02$ | $0.57 \pm 0.04$ | $0.34 \pm 0.02$ | $0.02 \pm 0.01$ |
| Disc. K-Means Transformer | 1 | $0.47$ | $0.42$ | $0.05$ | $0.0$ | $0.0$ | $0.0$ |
| *Diffusion Transformer | 3 | $3.74 \pm 0.02$ | $\mathbf{1.0 \pm 0.0}$ | $0.98 \pm 0.01$ | $0.9 \pm 0.02$ | $0.77 \pm 0.01$ | $\mathbf{0.1 \pm 0.01}$ |
| *Diffusion KDE Transformer | 3 | $\mathbf{3.93 \pm 0.02}$ | $\mathbf{1.0 \pm 0.0}$ | $\mathbf{1.0 \pm 0.0}$ | $\mathbf{0.95 \pm 0.0}$ | $\mathbf{0.89 \pm 0.01}$ | $0.1 \pm 0.01$ |
| *Diffusion Extra Transformer | 3 | $3.91 \pm 0.04$ | $\mathbf{1.0 \pm 0.0}$ | $0.99 \pm 0.01$ | $0.94 \pm 0.0$ | $0.88 \pm 0.01$ | $\mathbf{0.1 \pm 0.03}$ |
| | | | | | | | |
| Human | | $3.98$ | $1.0$ | $1.0$ | $1.0$ | $0.98$ | $0.0$ |
| | | | | | | | |
| Previously reported in (Shafiullah et al., 2022): | | | | | | | |
| Behaviour Transformers (K-Means+Residual) | 1 | $3.09$ | $0.99$ | $0.93$ | $0.71$ | $0.44$ | $0.02$ |
| Implicit BC (Generative EBM) | 1 | $2.71$ | $0.99$ | $0.87$ | $0.61$ | $0.24$ | $0.0$ |
| SPiRL VAE (Pertsch et al., 2021) | 1 | $1.0$ | $1.0$ | $0.0$ | $0.0$ | $0.0$ | $0.0$ |
| PARROT Normalizing Flow (Singh et al., 2020) | 1 | $0.04$ | $0.04$ | $0.0$ | $0.0$ | $0.0$ | $0.0$ |

Table 5: Wasserstein of time to completion in the robotic control environment. Methods marked with asterisk (*) are our proposed methods.

| | Task 1 ↓ | Task 2 ↓ | Task 3 ↓ | Task 4 ↓ |
|---|---|---|---|---|
| *Diffusion-Extra MLP Basic | $2.36 \pm 0.50$ | $3.65 \pm 0.26$ | $14.40 \pm 1.05$ | $14.04 \pm 0.38$ |
| *Diffusion MLP Basic | $3.48 \pm 0.94$ | $8.63 \pm 2.66$ | $23.13 \pm 4.86$ | $12.93 \pm 1.14$ |
| *Diffusion-KDE MLP Basic | $3.52 \pm 0.45$ | $5.82 \pm 0.40$ | $14.11 \pm 0.76$ | $8.86 \pm 0.74$ |
| | | | | |
| *Diffusion MLP Sieve | $2.69 \pm 0.97$ | $3.84 \pm 0.51$ | $8.54 \pm 2.81$ | $9.18 \pm 1.12$ |
| *Diffusion-KDE MLP Sieve | $3.50 \pm 0.68$ | $7.07 \pm 1.18$ | $8.51 \pm 2.08$ | $7.98 \pm 0.82$ |
| *Diffusion-Extra MLP Sieve | $1.69 \pm 0.26$ | $4.60 \pm 1.03$ | $8.34 \pm 2.36$ | $6.31 \pm 0.80$ |
| MSE MLP Sieve | $3.03 \pm 0.71$ | $5.70 \pm 1.89$ | $9.63 \pm 0.94$ | $7.24 \pm 0.91$ |
| Discrete MLP Sieve | $3.74 \pm 1.31$ | $11.09 \pm 1.92$ | $20.98 \pm 1.61$ | $9.38 \pm 2.12$ |
| K-Means MLP Sieve | $34.61 \pm 7.21$ | $71.20 \pm 20.31$ | - | - |
| K-Means+Residual MLP Sieve | $3.57 \pm 0.41$ | $10.07 \pm 2.96$ | $17.26 \pm 4.07$ | $15.51 \pm 1.36$ |
| | | | | |
| *Diffusion-Extra Transformer | $2.04 \pm 0.64$ | $5.12 \pm 1.02$ | $5.51 \pm 0.56$ | $5.92 \pm 0.31$ |
| *Diffusion Transformer | $\mathbf{1.26 \pm 0.01}$ | $\mathbf{3.33 \pm 0.36}$ | $7.89 \pm 0.57$ | $3.97 \pm 0.51$ |
| *Diffusion-KDE Transformer | $4.21 \pm 0.42$ | $6.34 \pm 1.23$ | $\mathbf{5.22 \pm 0.32}$ | $\mathbf{5.33 \pm 0.40}$ |
| MSE Transformer | $3.03 \pm 0.74$ | $5.73 \pm 0.61$ | $7.99 \pm 1.39$ | $6.64 \pm 1.12$ |
| Discrete Transformer | $2.35 \pm 0.31$ | $5.95 \pm 0.57$ | $9.61 \pm 1.82$ | $6.61 \pm 0.70$ |
| K-Means Transformer | $55.79$ | $32.64$ | - | - |
| K-Means+Residual Transformer | $2.16 \pm 0.78$ | $4.55 \pm 1.27$ | $10.07 \pm 0.23$ | $14.41 \pm 1.83$ |

Table 6: Timing analysis for various methods, sampling schemes, architectures, environments and hardware. Sampling times were recorded without running the environment – looping over the sampling process without waiting for the environment step function. Diffusion models do not suffer from any slow down at training time. For sampling time, note that in the Kitchen experiments, where the denoising network forms the majority of the total network, sampling speed is roughly proportional to number of denoising timesteps. However in CSGO, when a large observation encoder is required, the slow down is less impactful.

| | Training time per epoch ↓ | Sampling time ↓ | Max sampling rate ↑ |
|---|---|---|---|
| **Kitchen environment, CPU** | | | |
| MSE, MLP Sieve | 7.4 seconds | 1.2 ms | 833 Hz |
| Discrete, MLP Sieve | 7.6 seconds | 2.7 ms | 370 Hz |
| K-Means, MLP Sieve | 8.7 seconds | 1.5 ms | 666 Hz |
| K-Means+Residual, MLP Sieve | 10.5 seconds | 1.5 ms | 666 Hz |
| *Diffusion BC ($T = 50$), MLP Sieve | 6.4 seconds | 44.6 ms | 22 Hz |
| *Diffusion-X ($T = 50$, $M = 8$), MLP Sieve | 6.4 seconds | 55.9 ms | 18 Hz |
| *Diffusion-KDE ($T = 50$, 100 samples), MLP Sieve | 6.4 seconds | 128.4 ms | 8 Hz |
| **Kitchen environment, V100 GPU** | | | |
| MSE, Basic MLP | 1.4 seconds | 1.1 ms | 909 Hz |
| Discrete, Basic MLP | 1.8 seconds | 3.9 ms | 256 Hz |
| K-Means, Basic MLP | 2.2 seconds | 1.4 ms | 714 Hz |
| K-Means+Residual, Basic MLP | 2.4 seconds | 1.4 ms | 714 Hz |
| *Diffusion BC ($T = 50$), Basic MLP | 1.4 seconds | 42.1 ms | 24 Hz |
| *Diffusion-X ($T = 50$, $M = 8$), Basic MLP | 1.4 seconds | 47.3 ms | 21 Hz |
| *Diffusion-KDE ($T = 50$, 100 samples), Basic MLP | 1.4 seconds | 61.2 ms | 16 Hz |
| MSE, MLP Sieve | 1.5 seconds | 1.5 ms | 666 Hz |
| Discrete, MLP Sieve | 2.2 seconds | 4.3 ms | 232 Hz |
| K-Means, MLP Sieve | 2.9 seconds | 1.8 ms | 555 Hz |
| K-Means+Residual, MLP Sieve | 3.1 seconds | 2.0 ms | 500 Hz |
| EBM Deriv-Free, MLP Sieve | 22.3 seconds | 28.2 ms | 35 Hz |
| EBM Langevin, MLP Sieve | 142.1 seconds | 525.5 ms | 2 Hz |
| *Diffusion BC ($T = 50$), MLP Sieve | 2.0 seconds | 63.1 ms | 16 Hz |
| *Diffusion-X ($T = 50$, $M = 8$), MLP Sieve | 2.0 seconds | 73.8 ms | 14 Hz |
| *Diffusion-KDE ($T = 50$, 100 samples), MLP Sieve | 2.0 seconds | 86.5 ms | 12 Hz |
| MSE, Transformer | 6.8 seconds | 5.5 ms | 181 Hz |
| Discrete, Transformer | 7.0 seconds | 8.6 ms | 116 Hz |
| K-Means, Transformer | 7.4 seconds | 5.9 ms | 169 Hz |
| K-Means+Residual, Transformer | 7.5 seconds | 6.0 ms | 167 Hz |
| EBM Deriv-Free, Transformer | 1000< seconds | 1390.8 ms | 0.7 Hz |
| EBM Langevin, Transformer | 1000< seconds | 2927.9 ms | 0.3 Hz |
| *Diffusion BC ($T = 50$), Transformer | 6.8 seconds | 244.2 ms | 4 Hz |
| *Diffusion-X ($T = 50$, $M = 8$), Transformer | 6.8 seconds | 285.3 ms | 4 Hz |
| *Diffusion-KDE ($T = 50$, 100 samples), Transformer | 6.8 seconds | 295.6 ms | 3 Hz |
| **CSGO environment, ResNet18 observation encoder, V100 GPU** | | | |
| MSE, MLP Sieve | 49 seconds | 5.0 ms | 200 Hz |
| Discrete, MLP Sieve | 49 seconds | 6.0 ms | 167 Hz |
| EBM Deriv-Free, MLP Sieve | 50 seconds | 9.3 ms | 107 Hz |
| EBM Langevin, MLP Sieve | 240 seconds | 537.2 ms | 2 Hz |
| K-Means, MLP Sieve | 51 seconds | 5.5 ms | 181 Hz |
| K-Means+Residual, MLP Sieve | 51 seconds | 5.4 ms | 185 Hz |
| *Diffusion BC ($T = 20$), MLP Sieve | 49 seconds | 31.5 ms | 32 Hz |
| *Diffusion-X ($T = 20$, $M = 16$), MLP Sieve | 49 seconds | 54.7 ms | 18 Hz |

$t = 0$. Initialised state. Two modes can be made out in some action dimensions (e.g. dim 6), representing the choice between reaching for the kettle and microwave.

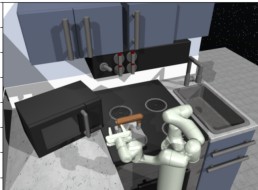 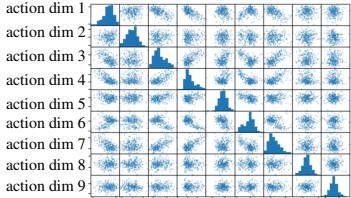

$t = 11$. The gripper begins moving towards the kettle and distributions become unimodal.

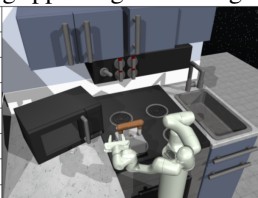 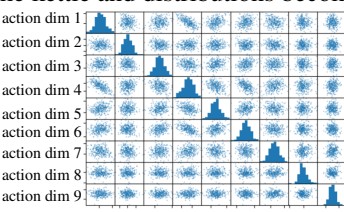

$t = 27$. The gripper initiates fine grasping of the kettle handle, which requires careful coordination between the grippers, which manifests as strong correlation between action dim 8 & 9.

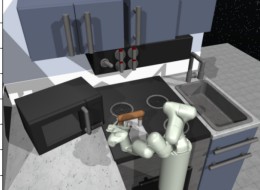 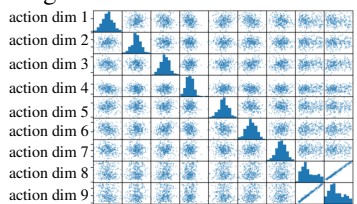

$t = 63$. Strong linear and non-linear correlations between certain action dimensions can be observed as the gripper pulls away from the kettle.

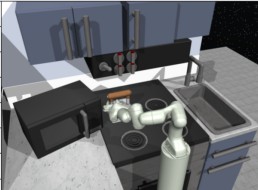 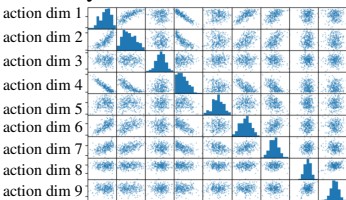

$t = 87$. Kettle task is completed. Bimodal distributions can be seen in action dimension 1 & 4, representing the decision point for the selection of the next task.

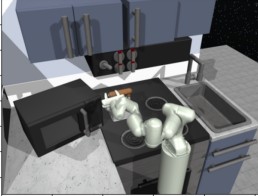 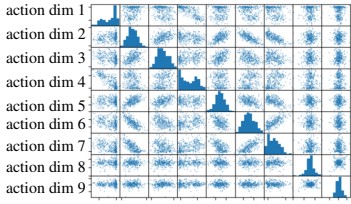

$t = 96$. The decision to move towards the bottom burner has been made, so action distributions become unimodal again.

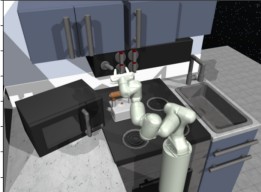 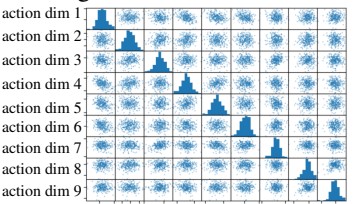

Figure 9: Qualitative analysis of a demonstrative roll-out for Diffusion BC in the kitchen environment. We visualise the relationships between all 9 action dimensions simultaneously via a matrix scatter plot, after 200 samples have been drawn from the diffusion model at each environment timestep. This allows visualisation of correlations and multimodality between action dimensions that are captured by the diffusion model. We have hand-selected six interesting environment timesteps in the roll-out to visualise. Action dimensions correspond to specific joint actuators of the robot, e.g. turn base left/right, or close gripper.

## D  SAMPLING ALGORITHMS

---

**Algorithm 1** Sampling for Diffusion BC

---

1: $\mathbf{a}_T \sim \mathcal{N}(\mathbf{0}, \mathbf{I})$
2: **for** $\tau = T, \dots, 1$ **do**
3:   $\mathbf{z} \sim \mathcal{N}(\mathbf{0}, \mathbf{I})$ if $\tau > 1$, else $\mathbf{z} = \mathbf{0}$
4:   $\mathbf{a}_{\tau-1} = \frac{1}{\sqrt{\alpha_\tau}} \left( \mathbf{a}_\tau - \frac{1-\alpha_\tau}{\sqrt{1-\bar{\alpha}_\tau}} \epsilon_\theta(\mathbf{a}_\tau, \tau, \mathbf{o}) \right) + \sigma_\tau \mathbf{z}$
5: **end for**
6: **return** $\mathbf{a}_0$

---

**Algorithm 2** Sampling for Diffusion-X

---

1: $\mathbf{a}_T \sim \mathcal{N}(\mathbf{0}, \mathbf{I})$
2: **for** $i = T, \dots, 1 - M$ **do**
3:   $\tau = \max(i, 1)$
4:   $\mathbf{z} \sim \mathcal{N}(\mathbf{0}, \mathbf{I})$ if $\tau > 1$, else $\mathbf{z} = \mathbf{0}$
5:   $\mathbf{a}_{\tau-1} = \frac{1}{\sqrt{\alpha_\tau}} \left( \mathbf{a}_\tau - \frac{1-\alpha_\tau}{\sqrt{1-\bar{\alpha}_\tau}} \epsilon_\theta(\mathbf{a}_\tau, \tau, \mathbf{o}) \right) + \sigma_\tau \mathbf{z}$
6: **end for**
7: **return** $\mathbf{a}_{-M}$

---

**Algorithm 3** Sampling for Diffusion-KDE

---

1: $\mathbf{A} \leftarrow [\,]$
2: **for** $i = 1, \dots, K$ **do**
3:   Use Algorithm 1 to sample action, $\mathbf{a}_0$.
4:   $\mathbf{A}$.append($\mathbf{a}_0$)
5: **end for**
6: KDEmodel.fit($\mathbf{A}$)
7: Likelihoods = KDEmodel.score($\mathbf{A}$)
8: $i = \arg\max_i(\text{Likelihoods})$
9: **return** $\mathbf{A}[i]$

---

## E  CLASSIFIER-FREE GUIDANCE ANALYSIS

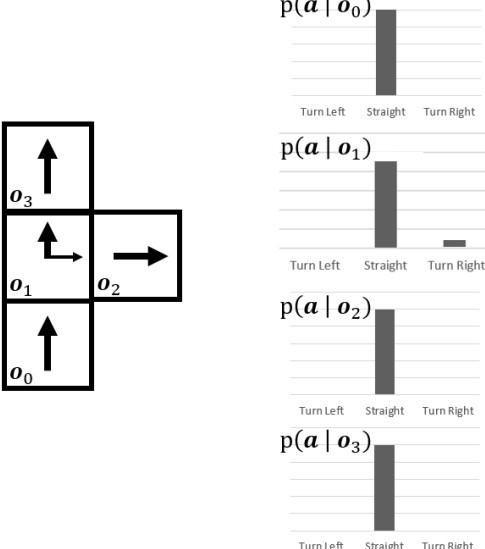

Figure 10: Didactic example of when CFG can lead to generation of less usual trajectories.

CFG can be interpreted as guiding the denoising sampling procedure towards higher values of $p(\mathbf{o}|\mathbf{a})$ (Ho et al., 2022). Given this interpretation, we now provide a grid-world to show concretely why this can lead to sampling of less common trajectories in a sequential environment.

Figure 10 shows an environment with four discrete states and a discrete action. The action space allows three ego-centric options; turn left, turn right or continue straight forward. Agents are always initialised at state 0, giving observation $\mathbf{o}_0$, and rolled out for exactly two timesteps, visiting state 1 and ending in state 2 or 3. The figure shows the empirical action distributions in a demonstration dataset. In states 0, 2 & 3, the agent always selects straight. But in state 1, the agent makes a right turn with 0.1 probability.

Let $\mathbf{o}_1$, $\mathbf{o}_2$ and $\mathbf{o}_3$ denote the observations given by states 1, 2 and 3, respectively. We can apply Bayes rule to find $p(\mathbf{o}|\mathbf{a})$, which will provide an understanding of what behaviour CFG induces. We are interested in the learnt behaviour at the decision point $\mathbf{o}_1$,

$$p(\mathbf{o}_1|\mathbf{a}) = \frac{p(\mathbf{a}|\mathbf{o}_1)p(\mathbf{o}_1)}{p(\mathbf{a})}. \tag{4}$$

Given the agent starts in state 0 and is rolled out for two timesteps (sees three states), and $p(\mathbf{a} = \text{Turn right}|\mathbf{o}_1) = 0.1$, $p(\mathbf{a} = \text{straight}|\mathbf{o}_1) = 0.9$, we find the marginal probability of each observation,

$$p(\mathbf{o} = \mathbf{o}_0) = 1/3 \tag{5}$$
$$p(\mathbf{o} = \mathbf{o}_1) = 1/3 \tag{6}$$
$$p(\mathbf{o} = \mathbf{o}_2) = 1/3 \cdot 0.1 \tag{7}$$
$$p(\mathbf{o} = \mathbf{o}_3) = 1/3 \cdot 0.9. \tag{8}$$

Hence, the marginal action distribution is,

$$p(\mathbf{a} = \text{Turn right}) = \sum_{i=0}^{3} p(\mathbf{a} = \text{Turn right}|\mathbf{o}_i)p(\mathbf{o}_i) \tag{9}$$

$$= 1/3 \cdot 0.1 \tag{10}$$

$$p(\mathbf{a} = \text{Straight}) = 1/3 + 1/3 \cdot 0.9 + 1/3. \tag{11}$$

We can now compute the quantities $p(\mathbf{o}_1|\mathbf{a})$,

$$p(\mathbf{o}_1|\mathbf{a} = \text{Turn right}) = \frac{p(\mathbf{a} = \text{Turn right}|\mathbf{o}_1)p(\mathbf{o}_1)}{p(\mathbf{a} = \text{Turn right})} = \frac{0.1 \cdot 1/3}{0.1 \cdot 1/3} = 1 \tag{12}$$

$$p(\mathbf{o}_1|\mathbf{a} = \text{Straight}) = \frac{p(\mathbf{a} = \text{Straight}|\mathbf{o}_1)p(\mathbf{o}_1)}{p(\mathbf{a} = \text{Straight})} = \frac{0.9 \cdot 1/3}{1/3 + 1/3 \cdot 0.9 + 1/3} = \frac{0.9}{2.9} = 0.31. \tag{13}$$

As such, since CFG favours actions that maximise $p(\mathbf{o}|\mathbf{a})$, the CFG agent will select the less frequently visited right-hand path more often.

