# OpenReview forum: "Imitating Human Behaviour with Diffusion Models"
_ICLR.cc/2023/Conference — ICLR 2023 poster_

### Official Review · Reviewer_hanC · 2022-10-20

**Confidence:** 3
**Correctness:** 3
**Technical Novelty And Significance:** 3
**Empirical Novelty And Significance:** 3
**Recommendation:** 8

**Clarity, Quality, Novelty And Reproducibility:**

- This paper is well written.
- Using diffusion models for imitation learning is new, and the proposed method is shown to be effective.
- Open-source code is promised.


###### Minor questions
- In Alg. 3, are some lines of $x_0$ missing?
- Diffusion-X is shown to be effective. It would be great to know the intuition behind its design.
- Does the "observation history" mentioned in this paper contain "o, t, a" or only the "o"?



**Strength And Weaknesses:**

###### Pros
- This paper provides (as far as the reviewer knows) one of the first detailed and substantially improved research on applying diffusion models to imitation human demonstrations.
- The toy experiments and discussions with the arcade claw game (with the figures) provide clear comparisons and convincing motivations.
- The proposed modifications to standard diffusion models are reasonable and shown to be suitable for sequential environments.
- The experiments of the kitchen environment with complex joint action space are convincing.

###### Cons
- The authors compared several methods (i.e., MLP, MLP Sieve, and Transformer), but these are more like ablated models. To demonstrate the superiority of diffusion models for observation-to-action modeling, comparisons to other generative models like GAN and flow models are helpful. Other state-of-the-art methods, like parrot [1] using realNVP for imitation, could also be considered.
- CSGO experiment is catchy but less convincing than the rest. The results for "16 x timesteps" and "32 x timesteps" metrics indicate all compared methods are far from satisfactory. More analyses and more baselines should be given for Table 3.
- Metrics like training time, evaluation time, or GPU consumption during training could better demonstrate the superiority of the proposed method.
- Does CFG not work due to the lack of data? Compared to text-to-image tasks with massive data, the observation-to-action task only has 566 trajectories with 25 task sequences. The conditions and reasons for CFG not working should be discussed rather than simply declaring that CFG is unsuitable for these tasks.

**Summary Of The Paper:**

This paper proposes to use diffusion models to imitate human behaviors. To make text-to-image model suitable for observation-to-action prediction, the authors introduce several modifications, including designing different architectures (MLP with embedding + skip connection, and  Transformer-based), removing Classifier-Free Guidance (CFG), and developing reliable sampling schemes. Experiments of task execution in a kitchen environment and playing CSGO show the effectiveness of the proposed method.

----
_After rebuttal_: The authors have moderately answered the questions and provided additional experiments. Besides, it seems that no critical issues were raised by other reviewers. Given the above, the reviewer would keep the original score (8).

**Summary Of The Review:**

- Overall, this is a well-written paper with clear ideas and good designs.
- More evaluation metrics and comparison methods could be added.

---

> ### Author Response · Authors · 2022-11-18
> **Author Response to Reviewer hanC**
>
> Thank you for your positive review, we’re pleased we were able to communicate the value of our investigation. We respond to your comments below.
>
> __Comment 4.1__ – Compare to other generative models
>
> Thank you for this suggestion, as this was also mentioned by Reviewer zdNu (see Comment 2.1). Our experimental set up is consistent with BeT, which allows us to compare to the baselines reported there – these include a VAE, a PARROT flow-based model, and implicit BC, which is a generative energy-based model (EBM) developed for BC. We have added all these lines to our Table 1 & 4. In addition, we re-implement an EBM into our code base – we have shared results and discussion in Appendix B.4.
>
> We see comparison to further generative models, such as GANs, as an interesting direction for further work, but since these have not (to our knowledge) previously been proposed and adapted for offline imitation learning (GAIL uses them to do online distribution-matching), we focused our effort on development and ablations of diffusion models.
>
> __Comment 4.2__ – Further experiments in CSGO
>
> Whilst the Kitchen is a lightweight environment allowing rigorous comparison of baselines and ablations, CSGO is larger in scale, requiring longer training and evaluation pipelines which makes it difficult to experiment to the same degree. Nevertheless, at your suggestion we have added two baselines, K-means+residual (the core component of BeT) and an EBM.
>
> In terms of metrics, the CSGO only contains the image as an observation so we are unable to do Wasserstein matching over states as was done in the kitchen environment, hence we did them over actions. Let us know if you’d like to see any further specific analysis?
>
> __Comment 4.3__ – Timing analysis
>
> We agree that a timing analysis would be useful for readers. We have added an analysis of training time and inference time in Table 6. See also Comment 2.4 to by Reviewer zdNu.
>
> __Comment 4.4__ – CFG failure
>
> We don’t believe the smaller dataset size is the cause of CFG’s failure. We have edited Section 3.3 and rewritten Appendix E to be more clear in our explanation.
>
> __Comment 4.5__ – Minor queries
>
> - Modified Alg. 3.
> - Added a sentence in 3.4 intuiting Diffusion-X (in essence we continue to move samples toward higher-likelihood regions).
> - Clarified that observation history only refers to “o’s” (the diffusion process is Markovian so there should be no benefit from adding previous noised actions).

---

### Official Review · Reviewer_Ygtk · 2022-10-23

**Confidence:** 3
**Correctness:** 3
**Technical Novelty And Significance:** 2
**Empirical Novelty And Significance:** 3
**Recommendation:** 6

**Clarity, Quality, Novelty And Reproducibility:**

- Besides some notation issues, the paper is mostly well-written and easy to follow.
- The paper has good conceptual novelty as it proposes to model the policy with diffusion models to capture multi-modal behaviors. The technical novelty is not huge for the network architecture.
- The authors promise to open source the code, which should make reproducing the work relatively easy.

**Strength And Weaknesses:**

**Strength:**

- The motivation for using diffusion models for imitation learning is illustrated well with the toy task, where the limitation of standard action models is clear.
- The proposed sampling methods are effective in improving task completion. The paper also shows that classifier-free guidance is harmful to policy modeling, both through experiments and didactic examples.
- Experiments are performed on semi-realistic environments such as simulated robotic control and photorealistic video games, which clearly demonstrated the effectiveness of the proposed approach. Metrics are also comprehensive, evaluating both the quality and coverage of the produced actions.

**Weakness:**

- The paper listed the network architecture as the first contribution, but it’s similar to prior work on diffusion models in essence and appears incremental.
- It would be great to visualize the learned diffusion policies and demonstrate different modes of action produced by the policy (for robotic control and CSGO), which seems to be the main purpose of using diffusion models.
- Some notations are confusing, especially the use of $t$ since it normally denotes the RL timesteps. I would recommend using another symbol for the diffusion process.

**Summary Of The Paper:**

The paper proposes to use diffusion models to model human behavior imitation policies learned with behavior cloning. The motivation is that human behaviors can be complex and multi-modal, and simple Gaussian or categorical policies are unable to capture the underlying multi-modal distributions. The paper utilizes DDPM as the diffusion process and proposes simple architectures to denoise the action distribution. A stable sampling scheme is also proposed to avoid bad actions. Experiments are performed on robot control tasks and first-person shooting games, which demonstrate that the method outperforms the baseline action distribution models across various metrics and achieves better coverage of demonstrations. The paper also discussed and showed through experiments that classifier-free guidance is not suitable for modeling the policy, since it may blend in actions from unrelated states.

**Summary Of The Review:**

I appreciate the idea of using diffusion models for imitation learning since they can capture multi-modal behaviors better. The technical novelty is not huge, but the experiments are extensive and the method can serve as a good baseline for diffusion-based imitation learning. I’m leaning toward acceptance at this point.

---

> ### Author Response · Authors · 2022-11-18
> **Author Response to Reviewer Ygtk**
>
> Many thanks for your thoughtful review, we are pleased that you have seen particular value in our experiments. We respond to your comments below.
>
> __Comment 3.1__ – Visualisation of diffusion policies
>
> Appendix Figure 7 visualises a scatter matrix of the 9 action dimensions in the kitchen environment for one roll out. We have selected several points in the trajectory, some of which display correlations or multimodality, demonstrating that the flexibility of the diffusion model is indeed important in this environment. We have updated the figure caption to be clearer in what it portrays, apologies for not emphasizing this originally.
>
> __Comment 3.2__ – Technical novelty
>
> We believe it’s the _simplicity_ of observation-to-action diffusion that is its strength. While the core method might be simple, we note that Reviewer hanC commented our investigation is "one of the first detailed and substantially improved research on applying diffusion models to imitating human demonstrations." We hope that our technical contributions in new sampling schemes and insights around CFG will be valued by the ICLR community.
>
> __Comment 3.3__ – Notation
>
> Thank you for pointing this out. We have replaced instances of $t$ with $\tau$ and renamed ‘timesteps’ as ‘environment timesteps’ or ‘diffusion timesteps’ as applicable.

---

### Official Review · Reviewer_zdNu · 2022-10-24

**Confidence:** 4
**Correctness:** 3
**Technical Novelty And Significance:** 2
**Empirical Novelty And Significance:** 2
**Recommendation:** 6

**Clarity, Quality, Novelty And Reproducibility:**

The paper is very well-written and easy to follow. The paper introduces and adapts diffusion models to behavioral cloning for a range of simulated environments. Employing diffusion models for observation-to-action imitation learning along with the proposed adaptations can be considered novel. Enough details are provided to reproduce the discussed method.

Detailed comments:
- It is not clear what is shown in Figure 3. What do the columns represent?
- In the Second paragraph in Section 3.4 the text refers to 'human demonstrations'. As stated above, it is not clear what these are and how they were captured. A discussion about the details of the used datasets would be helpful.
- In Figure 4 examples of 'Diffusion-X' and 'Diffusion-KDE' are shown. The text referring to the figure in Section 3.4 also discusses 'Diffusion BC'. I would appreciate a visualization of 'Diffusion BC'.
- For the robotic control and counter strike experiments it would beneficial to more carefully report the inference time of the diffusion model. Section 4.2. mentions that a 'run rate of around 10Hz' was measured, however, a more detailed analysis would provide additional insights, also for future methods that may want to compare to the proposed approach. This is supposed to be in addition to the task runtime provided in Table 5.
- What exactly is shown in the right plots in Figure 7? What is a certain point in the rollout?
- It is not clear what is discussed in Appendix Section E. More details should be provided for this section.


**Details Of Ethics Concerns:**

Behavioral cloning is an established research direction in the field of robotics. The experiments discussed in this paper only center around simulated data and rather constraint environments. Therefore, I do not have any immediate concerns regarding the discussed approach.


**Strength And Weaknesses:**

Strengths:

- The paper addresses an important problem with a sound technical solution.
- The paper is very well written and easy to read.
- The paper discusses a set of meaningful experiments that show the benefits of employing diffusion models.
- The comparison of different modeling choices provides an insightful summary.

Weaknesses:

- The discussed baselines (MSE, Gaussian, etc.) for behavior cloning appear rather weak and a comparison to other generative approaches should be provided.
- The paper makes too strong claims towards modeling 'human behaviour' that should be toned down.
- The paper does not discuss any limitations.


**Summary Of The Paper:**

This paper studies the use of diffusion models as observation-to-action models in the context of behavioral cloning. In particular, different version of DMs are explored for three different environments (claw arcade game, robotic control, counter strike). The experiments are technically sound and employing diffusion models for imitations learning shows promising results.

**Summary Of The Review:**

Overall, this is an interesting paper that can be accepted to ICLR 2023. The paper is very well written, the experiments are sound, and the problem the proposed method addressees is important. Using diffusion models for behavioral cloning is an interesting and new direction and the paper also has the potential to stimulate future work in this direction.

However, there is also a number of shortcomings that I would like to see addressed prior to acceptance. For one, I am missing a comparison to other generative approaches as baselines. I was somewhat surprised that no other learning-based approach was used as a baseline for the arcade claw game experiment. What is the reason for that? Second, for the claw experiment the paper fails to provide a discussion on how human demonstrations where captured and what exactly the action space is. The appendix states that samples were drawn from the true p(a|o). So here no human demonstrations where used at all?

Given the paper only discusses environments with rather constraint action spaces, I find that the title 'Imitating Human Behaviour' is quite misleading and should be changed (e.g. to 'Imitating Human Demonstrations'). That the method is able to imitate human behavior arguably is a too strong statement that should be toned down throughout the paper (e.g. in Section 5).

---

> ### Author Response · Authors · 2022-11-18
> **Author Response to Reviewer zdNu**
>
> Thank you for your detailed comments. We have incorporated your feedback into a new version of the paper. We hope these changes might justify an increase in your score.
>
> __Comment 2.1__ – Weak baselines
>
> Two SOTA BC methods are behaviour transformer (BeT) [1], and implicit BC (IBC) [2]. We currently compare to BeT both through the numbers reported by [1] and via our baseline named “K-means+Residual”, which reimplements the core innovation of BeT, but standardizes the architecture and training details to be in line with other methods. We have made this clearer in Section 4’s baseline paragraph.
>
> IBC is a generative model implemented with an energy-based model (EBM). As far as we know this is SOTA in generative models applied to imitation learning. Note that [1] already ran the official IBC implementation in Kitchen, which allows us to directly compare with it in Table 1. We have added their reported results for a flow-based model and a VAE in Table 1 & 4.
>
> We agree that incorporating a generative EBM as a baseline into our own code base would be a valuable comparison for readers. We have spent considerable time reimplementing two versions of this generative EBM: 1) Derivative-free 2) Langevin MCMC. We provide discussion on the method in Appendix B.4. Note that Figure 8 now contains this learning-based method visualised on the Claw environment.
>
> __Comment 2.2__ – Modelling human behaviour claims
>
> The core motivation for our investigation was that the flexibility of diffusion models would be advantageous when replicating human action distributions. Note that little benefit would be gained from this flexibility if the demonstration policy was itself of limited expressiveness. Saying that, we are happy to follow your advice that our claims are too strong in this direction and have edited the paper in several places:
> - Section 2. “they are an excellent fit for imitating human behaviour” -> “they provide benefits when imitating human demonstrations”
> - Section 4. “imitate human behaviour” -> “imitate human demonstrations,”
> - Section 4.1.1. “human behaviour”-> “human demonstrations”
> - Section 4.2 title. “Modelling Human Players”-> “Modelling Human Gameplay”
> - Last paragraph of Section 5.
> - We are willing to revise the paper title as per our summary response if all reviewers and the AC agree in their final recommendations.
>
> __Comment 2.3__ – Limitations
>
> We have added a paragraph summarising limitations (e.g. increased inference time) of the approach in Section 5.
>
> __Comment 2.4__ –  Run-time analysis
>
> We have added analysis into training and inference time for various models and architectures in Table 6.
>
> __Comment 2.5__ –  Claw experiment data
>
> The Claw was intended as a toy example to communicate intuitions about the methods. The data was synthetically generated and did not use human demonstrators. The action space was continuous 2D. We have clarified this at the start of Section 2. Learning-based EBM baselines are now compared against in Figure 8.
>
> __Comment 2.6__ – Detailed comments
>
> We have addressed all of these in the revised version, updating figure captions and rewriting Appendix E.
>
> __References__
>
> [1] Behavior Transformers: Cloning k modes with one stone, Shafiullah et al. 2022
>
> [2] Implicit Behavioral Cloning, Florence et al. 2021

---

### Official Review · Reviewer_PK6N · 2022-10-25

**Confidence:** 3
**Correctness:** 3
**Technical Novelty And Significance:** 3
**Empirical Novelty And Significance:** 3
**Recommendation:** 8

**Clarity, Quality, Novelty And Reproducibility:**

Clarity - very clear, minor details and reasoning could be improved.
Quality - high quality
Novelty - relatively novel
Reproducibility - unsure

**Strength And Weaknesses:**

Strengths:
- Well written, and arguments well supported
- Detailed results, varying baselines implemented for tasks
- Core idea is relatively novel and has a lot of potential

Weaknesses:
- Some implementation details could be improved.
- Key insight - " difussion models avoid BC shortcomings by making no model assumptions", doesn't seem well substantiated from the results as stated.

**Summary Of The Paper:**

This paper attempts to use diffusion models for action generation in imitation learning set ups. The paper contributes :
1. modifications to diffusion models to adapt them to action generation
2. discussion on effectiveness of CFG
3. sampling techniques for action generation
The paper tests on a few datasets including robot manipulation in kitchen and video games.

**Summary Of The Review:**

This paper presents a novel idea and is well written. Discussion on how key insights can be derived from experimental results can be improved.

---

> ### Author Response · Authors · 2022-11-18
> **Author Response to Reviewer PK6N**
>
> Thank you for your comments. We are pleased to have been able to communicate the value of our work. We respond to your comments below.
>
> __Comment 1.1__ – Implementation details could be improved
>
> We aimed to be comprehensive in our description of our implementation and hyperparameters in Appendix B. We are happy to add any further specific details you feel may be helpful? Note that we will upload experiment code if the paper is accepted to provide full transparency over our implementation.
>
> __Comment 1.2__ – “No assumptions” claim
>
> We have reworded our claims to be of the form “diffusion models make no coarse assumptions about the action distribution, so avoid some of the shortcomings of other methods”, which we believe to be an honest reflection of the model. Please let us know if you’d like further modification.
>
> __Comment 1.3__ – Discussion on how key insights can be derived from experimental results can be improved
>
> We have edited our conclusion (Section 5, third paragraph) to incorporate these findings explicitly.

---

### Author Response · Authors · 2022-11-18
**Author Response to All Reviewers**

We would like to thank all of the reviewers for their hard work and constructive feedback. Several points were raised which have helped guide us in further strengthening the paper. While we respond to each reviewer directly, we summarise several improvements here.

- __Model baselines.__ We have reimplemented a generative energy-based model (aka Implicit BC) into our own code base (details in Appendix B.4). We added several strong generative baselines run in prior work on the Kitchen environment. We have revised our description of other baselines in Section 4.
- __Timing analysis.__ We have documented training and sampling time for various models, hardware and environments in Table 6.
- __Wording changes.__ We have edited the text in numerous places to tighten up claims and clarify descriptions.
- __Title change.__ Reviewer zdNu, suggested a change to “Imitating human demonstrations with diffusion models”. We are fine with this, but will await on a consensus from reviewers and AC before updating this on the submission page.

We are available to respond to further queries as needed.

---

### Decision · Program_Chairs · 2023-01-20

**Decision:**

Accept: poster

**Justification For Why Not Higher Score:**

Somewhat limited novelty (comes across as nearly off-the-shelf kind of application) and while the experiments are sound, baselines could have been stronger to merit a spotlight.

**Justification For Why Not Lower Score:**

Reasonable, well executed approach documenting a useful diffusion-based baseline for the imitation learning literature. No reason to reject given positive reviews.

**Metareview: Summary, Strengths And Weaknesses:**

Summary: An application of diffusion models for learning observation-conditioned action distributions for imitation learning. Proposes diffusion architectures for action denoising, a stable sampling scheme to avoid bad actions; and shows that classifier-free guidance is not suitable for modeling the policy, since it may blend in actions from unrelated states.

Strengths: Well written and well motivated. Experiments  on simulated robotic control and photorealistic video games show favorable results.

Weaknesses: The technical novelty is limited, but the experiments are extensive and the method can serve as a good baseline for diffusion-based imitation learning. Baselines (MSE, Gaussian, etc.) for behavior cloning appear rather weak and a comparison to other generative approaches should be provided.

**Note From Pc:**

if the above contains the word "oral" or "spotlight" please see: "oral" presentation means -> notable-top-5% and "spotlight" means -> notable-top-25%. As stated in our emails, we are disassociating presentation type from AC recommendations